# Spectroscopic visualization of reversible hydrogen spillover between palladium and metal–organic frameworks toward catalytic semihydrogenation

Qiaoxi Liu[1,2], Wenjie Xu[1], Hao Huang[1], Hongwei Shou[1], Jingxiang Low [1], Yitao Dai[1], Wanbing Gong[1], Youyou Li[1], Delong Duan[1], Wenqing Zhang[1], Yawen Jiang[1], Guikai Zhang[3], Dengfeng Cao[1], Kecheng Wei[1], Ran Long [1] ✉, Shuangming Chen [1], Li Song [1] & Yujie Xiong [1,2,4] ✉

Hydrogen spillover widely occurs in a variety of hydrogen-involved chemical and physical processes. Recently, metal–organic frameworks have been extensively explored for their integration with noble metals toward various hydrogen-related applications, however, the hydrogen spillover in metal/MOF composite structures remains largely elusive given the challenges of collecting direct evidence due to system complexity. Here we show an elaborate strategy of modular signal amplification to decouple the behavior of hydrogen spillover in each functional regime, enabling spectroscopic visualization for interfacial dynamic processes. Remarkably, we successfully depict a full picture for dynamic replenishment of surface hydrogen atoms under interfacial hydrogen spillover by quick-scanning extended X-ray absorption fine structure, in situ surface-enhanced Raman spectroscopy and ab initio molecular dynamics calculation. With interfacial hydrogen spillover, Pd/ZIF-8 catalyst shows unique alkyne semihydrogenation activity and selectivity for alkynes molecules. The methodology demonstrated in this study also provides a basis for further exploration of interfacial species migration.

Hydrogen spillover is a highly important process governing the performance of various hydrogen-related chemical and physical applications such as hydrogen embrittlement[1–3], atomic metal hydrogen[4,5], hydrogen storage[6,7], hydrogen sensing[8], and hydrogen-involved catalysis[9–11]. In fact, the research on hydrogen spillover has been a long journey since its first discovery on WO₃/Pt system[12], during which some studies have proven the existence of hydrogen

migration on metal[13] and metal oxide surfaces[14]. Traditionally, metal components are often supported on metal oxides for hydrogen-related applications. To deepen fundamental understanding, the interfacial hydrogen spillover in metal/metal oxide systems has been measured by developing characterization techniques[15,16]. Nevertheless, the application-driven materials design has evolved to more complex levels—integration of metal nanoparticles with soft

[1]Hefei National Research Center for Physical Sciences at the Microscale, Key Laboratory of Precision and Intelligent Chemistry, School of Chemistry and Materials Science, School of Nuclear Science and Technology, National Synchrotron Radiation Laboratory, University of Science and Technology of China, Hefei, Anhui 230026, China. [2]Suzhou Institute for Advanced Research, University of Science and Technology of China, Suzhou, Jiangsu 215123, China. [3]Beijing Synchrotron Radiation Facility, Institute of High Energy Physics, Chinese Academy of Sciences, Beijing 100049, China. [4]Key Laboratory of Functional Molecular Solids, Ministry of Education, Anhui Engineering Research Center of Carbon Neutrality, College of Chemistry and Materials Science, Anhui Normal University, Wuhu, Anhui 241000, China. ✉e-mail: longran@ustc.edu.cn; yjxiong@ustc.edu.cn

materials. A typical example is the vast development of heterogeneous metal/MOF composite structures. MOFs are crystalline porous hybrid materials composed of metal nodes and organic linkers via strong covalent bonds to form extended networks, which offer outstanding features, including high internal surface area, large porosity, structural tunability, and potentially high density of active sites, to enhance the performance of metal nanoparticles. Indeed, metal/MOF composite structures have been widely employed in numerous hydrogen-related applications such as selective hydrogenation[17–20], dehydrogenation[21,22], and hydrogen storage[23,24]. While most studies ascribed such unique properties of metal/MOF structures to electronic effects or steric effects, hydrogen spillover through the metal−MOF interface is an underlying factor that should have a potential influence on overall performance. Along with the demonstration of hydrogen spillover in MOFs, it has been proposed that hydrogen spillover may occur in metal/MOF structures[23,25–27].

Despite the prediction, direct evidence for the dynamic behavior of hydrogen atoms in metal/MOF systems is lacking[28], which has largely limited the understanding and further enhancement of their performance in hydrogen-related applications. The successful characterization of metal/metal oxide systems is incentive to the case of metal/MOF structures, yet the real situation involving MOFs is significantly more complex. Here we take Pd−a metal uniquely interacting with hydrogen atoms−as an example to highlight the system complexity. In Pd/MOF structures, tracking hydrogen spillover pathways is greatly challenged by the presence of organic linkers and metal nodes in MOFs, not to mention that hydrogen atoms may migrate into MOF or Pd bulk lattices forming complex dynamic processes. In the case of pristine Pd (Fig. 1a), $H_2$ molecule is nearly spontaneously dissociated into hydrogen atoms ($H_{ad}$) on Pd surface, which can then diffuse into bulk lattice of Pd in addition to migrate on surface[29,30]. Given the simplicity of pristine Pd, it has been elucidated that the bulk lattice of Pd becomes saturated with $H_{ad}$ atoms, forming palladium hydrides ($PdH_x$). In fact, palladium has a face-centered cubic (fcc)

lattice whose octahedral sites can be partially or completely occupied by $H_{ad}$ atoms to form $PdH_\alpha$ and $PdH_\beta$, respectively. With increasing $H_{ad}$ concentration, the phase of palladium hydrides can change from $PdH_\alpha$ to $PdH_\beta$[31,32]. Depending on the $H_{ad}$ concentration, the $H_{ad}$ atoms may again migrate out of the bulk lattice[29,30]. When Pd is interfaced with MOF (Fig. 1b), hydrogen migration pathways would become elusive. While certainly the $H_{ad}$ atoms generated on Pd surface (i.e., Pd−MOF interface) can still diffuse into or out of Pd bulk lattice, it lacks direct evidence whether they may migrate into MOF or back to Pd−MOF interface. Once this process in MOF could take place, there is little doubt that the $H_{ad}$ concentration at the interface would be altered, impacting the process inside Pd bulk.

Intuitively, this mystery can be unraveled by directly examining MOF upon $H_2$ dissociation. Solid-state nuclear magnetic resonance (NMR) spectroscopy is a useful tool for looking into chemical environment of atoms in MOFs[33]. Figure 1c shows our attempt to NMR characterization of Pd/MOF structures exposed to $H_{ad}$ atoms. We employ Zn(2-methylimidazole)$_2$ (ZIF-8) as a model MOF (Supplementary Fig. 1). To exclude the hydrogen signal originating from ZIF-8 ligands, $D_2$ is used as a hydrogen source to investigate hydrogen spillover in Pd/ZIF-8 structures. In the case of pristine Pd, a peak is observed at 4.04 ppm, corresponding to the $^2H$ atoms inside Pd lattice from $D_2$ dissociation[34]. After interfacing with ZIF-8, a peak arises at 0.93 ppm in addition to the weakened peak for $^2H$ in Pd lattice (23.0 ppm). This emerging peak is located at higher field as compared with the peak position of pristine Pd. It has been reported that the H absorption line in Pd bulk is linearly shifted to lower field with increase of H concentration[35], resulting from the addition of MOF coating[23], in agreement with our observation at 23.0 ppm. While bare ZIF-8 does not show any $^2H$ peak, we suspect that this emerging peak at 0.93 ppm is related to the spillover of $^2H$ atoms into ZIF-8. However, NMR signals are susceptible to many other factors, and Pd/ZIF-8 is such a complex system involving Pd, Zn nodes, and organic ligands. As such, we can hardly reach a conclusion based on such a signal change in NMR[28]. In

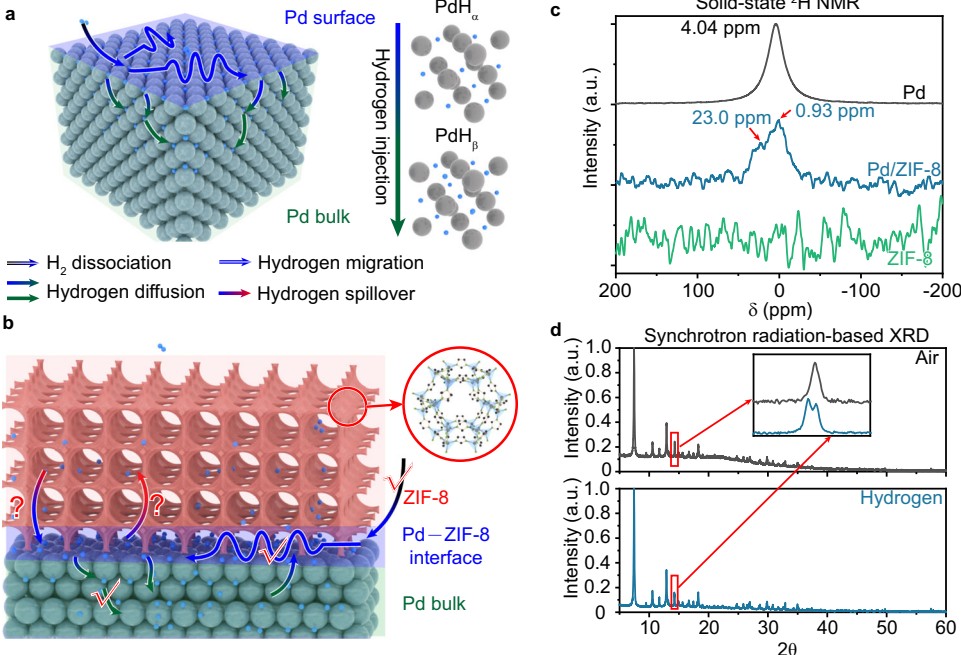

**Fig. 1 | Challenges of collecting direct evidence for hydrogen spillover in Pd/MOF structures. a** Schematic illustration of hydrogen migration in pristine Pd and formation of palladium hydrides as reported in literature. **b** Schematic illustration of potential hydrogen migration pathways in Pd/MOF structures. Pd atoms: green balls; hydrogen atoms: blue balls; ZIF-8 coating: pink porous structure. The color of arrows represents the start point and the destination of hydrogen migration.
**c** Solid-state $^2H$ NMR spectra for Pd/ZIF-8 structures exposed to $D_2$, in reference to ZIF-8 and Pd nanoparticles. **d** Synchrotron radiation-based XRD patterns of Pd/ZIF-8 structures before and after exposure to $H_2$. All data above have been normalized.

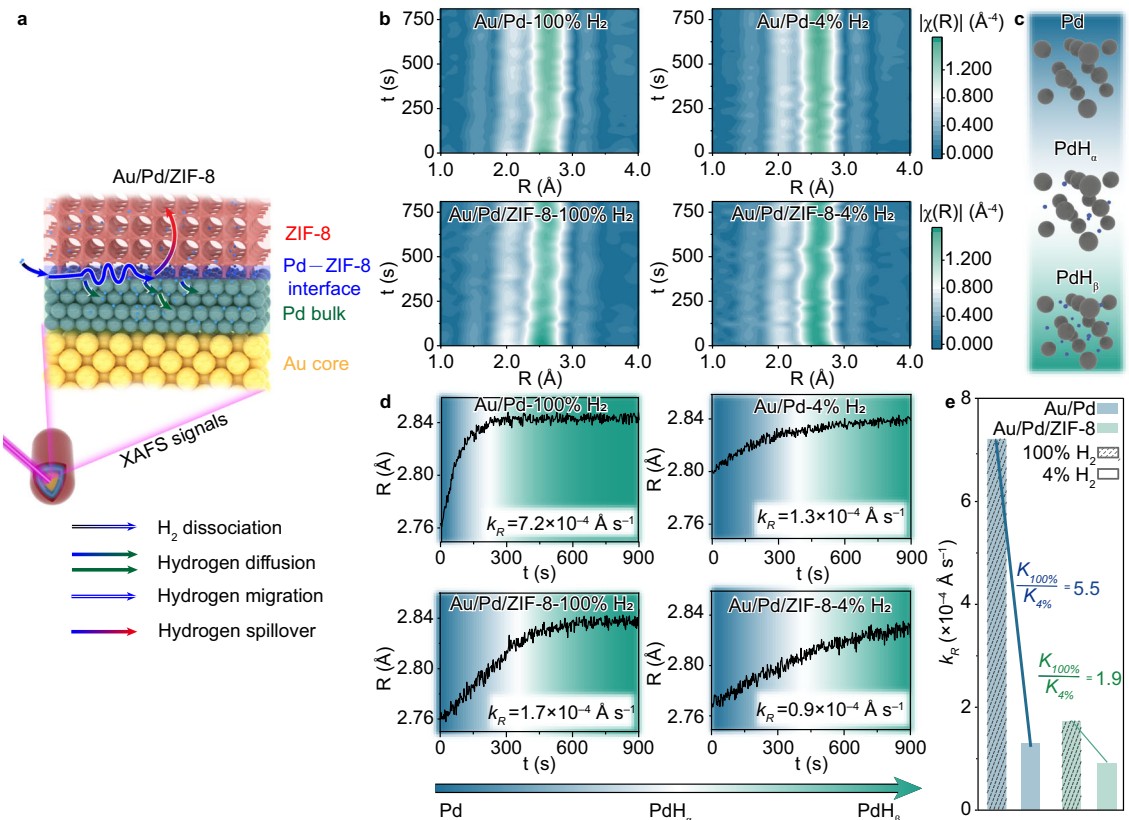

**Fig. 2 | Quick-EXAFS characterization of hydrogen spillover in interface−Pd bulk regime. a** Schematic illustration of Au/Pd/ZIF-8 model for quick-EXAFS characterization. Pd atoms: green balls; hydrogen atoms: blue balls; Au atoms: yellow balls; ZIF-8 coating: pink porous structure. The color of arrows represents the start point and the destination of hydrogen migration. **b** In situ Fourier transformed magnitudes of the $k^2$-weighted Pd K-edge EXAFS spectra of Au/Pd and Au/ Pd/ZIF-8 at 100% $H_2$ or 4% $H_2$. **c** The crystalline structure of Pd, $PdH_\alpha$, and $PdH_\beta$ (the black balls represent Pd atoms, and the blue balls represent H atoms). **d** Time-dependent bonding distance of Pd-Pd of Au/Pd and Au/Pd/ZIF-8 at 100% $H_2$ or 4% $H_2$. **e** The rates of Pd-Pd distance change ($k_R$) of Au/Pd and Au/Pd/ZIF-8 at different $H_2$ concentrations. The color transition regions correspond to Pd (blue), $PdH_\alpha$ (white), and $PdH_\beta$ (green) with different palladium hydride phases.

---

parallel, we can observe a peak shift for $PdH_\beta$ in commercial X-ray diffraction (XRD) (Supplementary Fig. 2) and a peak split for ZIF-8 in synchrotron radiation-based XRD (Fig. 1d, Supplementary Fig. 3), indicating the change in ZIF-8. Nonetheless, we still cannot conclude the process occurring in ZIF-8 frameworks. The grand challenge, which we have encountered in traditional characterizations, urges us to develop an approach for revealing the dynamic processes of $H_{ad}$ atoms from the starting point where the $H_{ad}$ atoms are generated−Pd−ZIF-8 interface.

Here, we apply modular spectroscopic characterizations which separately examine two different functional regimes (i.e., interface−Pd bulk regime and interface−ZIF-8 regime, Fig. 1b). Specifically, the characterizations require amplifying the related signals to decouple the contributions of ZIF-8 component, Pd component, and Pd−ZIF-8 interface. To this end, the third module, Au with the functions of signal initiation and signal amplification, is reasonably introduced into the metal/MOF composite structure. As such, we elaborately decouple the interface−Pd bulk regime and interface−ZIF-8 regime from the composite structure, respectively, which are closely examined via quick-scanning extended X-ray absorption fine structure (quick-EXAFS) and in situ surface-enhanced Raman spectroscopy (SERS) with two model structures: Au/Pd/ZIF-8 and Pd/ZIF-8/Au. We reveal the effect and existence of hydrogen spillover on Pd−ZIF-8 interface by spectroscopic results of quick-EXAFS and in situ SERS. Combined with ab initio molecular dynamics (AIMD) calculation, the full pathway of hydrogen spillover and the dynamic replenishment of surface hydrogen atoms are depicted. The Pd/ZIF-8 catalyst shows high hydrogenation activity and selectivity for various unsaturated compounds especially under

low $H_2$ concentration. This work demonstrates the potential of modular signal amplification in investigating interfacial species migration.

## Results and discussion
### Quick-EXAFS characterization
We first look into the interface−Pd bulk regime, which can evaluate the impact of ZIF-8 coating on hydrogen spillover and diffusion into Pd bulk, using Au/Pd/ZIF-8 as a model (Fig. 2a). In this modular construction, Au/Pd/ZIF-8 is a core−shell structure based on Au nanorods (Supplementary Fig. 4). The Au nanorods with an average diameter of *ca.* 16 nm and length of *ca.* 66 nm are first covered with Pd shells with a thickness of *ca.* 3 nm. Then, ZIF-8 is coated on Au/Pd with an edge length of *ca.* 250 nm, forming a Au/Pd/ZIF-8 structure. In such a structure, Pd enables nearly spontaneous dissociation of $H_2$ molecules, while Au nanorods do not take part in hydrogen spillover according to the forbiddance of hydrogen migration between Pd and Au[26,36]. In the meantime, the existence of Au core avoids the body dilution of near surface signal in X-ray absorption spectroscopy (XAS). Furthermore, the tunability of Pd shell, as well as the periodicity of metal−MOF interface across the structure, can significantly amplify quick-EXAFS signals, making the visualization of hydrogen spillover effect in the interface−Pd bulk regime possible. Leveraging the kinetics information for hydrogen diffusion into pristine Pd lattice[29–32], we can examine the changes of hydrogen spillover and diffusion with the addition of ZIF-8 coating.

To gather the information for hydrogen diffusion kinetics tuned by ZIF-8, we collect EXAFS signals at Pd K-edge of Au/Pd and Au/Pd/ ZIF-8 upon exposure to different hydrogen atmospheres. As such, the

dynamic evolution of Pd bulk lattice can be analyzed by collecting over 3600 spectra for each sample via quick-EXAFS. The nearly sponta-neous dissociation of $H_2$ on Pd surface makes the Pd bulk saturated with $H_{ad}$ to form $PdH_x$, which gradually evolves from $PdH_\alpha$ to $PdH_\beta$ with increase of $H_{ad}$ concentration[29-32]. The hydrogen atoms occupy the octahedral sites gradually and will reach the maximum occupation where the Pd evolves to $PdH_\beta$ entirely (Fig. 2c)[32]. By fitting the collected quick-EXAFS spectra (Supplementary Figs. 5-9), such an evolution of Pd lattice can be resolved through Pd–Pd interatomic distances (R). As shown in Fig. 2b–d, the Pd–Pd interatomic distances are gradually expanded, suggesting that hydrogen atoms begin entering the lattice of palladium. When Pd–Pd interatomic distances reach a constant value, the hydrogen atoms occupy maximum octahedral sites. This state was commonly defined as $PdH_\beta$. The state when Pd–Pd intera-tomic distances increase steadily was defined as $PdH_\alpha$. The turning point in the increase was defined as the mixed phase of $PdH_\alpha$ and $PdH_\beta$. Beyond a certain point of time, the Pd–Pd interatomic distances reach a constant value of 2.84 Å where the expansion rate of Pd–Pd intera-tomic distances is 2.90% due to the stabilization of palladium hydrides.

To better evaluate the effect of MOF on Pd hydrogenation, the rates of Pd–Pd distance change ($k_R$) from Pd to $PdH_\beta$ that represent the hydrogenation rate of Pd are summarized as Fig. 2d. Under 100% $H_2$ condition, the rate of Pd–Pd distance change ($k_R$) is $1.7 \times 10^{-4}$ Ås$^{-1}$ for Au/Pd/ZIF-8 and $7.2 \times 10^{-4}$ Ås$^{-1}$ for Au/Pd. As the atmosphere is swit-ched to 4% $H_2$, $k_R$ is lowered to $0.9 \times 10^{-4}$ Ås$^{-1}$ for Au/Pd/ZIF-8 and $1.3 \times 10^{-4}$ Ås$^{-1}$ for Au/Pd. Many factors, such as surface defect and charge transfer interaction, affect the hydrogenation rate. For Au/Pd and Au/Pd/ZIF-8, it is complicated to interpret the difference in $k_R$ of the two samples. To avoid the influence of chemical environment change by ZIF-8 coating, $k_R$ with different hydrogen concentrations can be directly compared. As the research by Borgschulte et al. proposed[29], the rate of Pd hydrogenation should be proportionally correlated to the applied pressure of $H_2$. It is acknowledged that the hydrogenation rate is proportional to surface hydrogen concentration according to the Fick's law of diffusion. The ratio of the rates at 100% $H_2$ and 4% $H_2$ for Au/Pd, $\frac{k_{R100\%}}{k_{R4\%}} = 5.5$, is substantially higher than that for Au/Pd/ZIF-8, $\frac{k_{R100\%}}{k_{R4\%}} = 1.9$ (Fig. 2e). This indicates that the Pd hydro-genation process for Au/Pd/ZIF-8 deviates from the theoretical law of Pd lattice that Au/Pd approximately fits. In particular, the ZIF-8 layer in Au/Pd/ZIF-8 enables an extraordinarily insensitive property for $H_2$ concentration. For Au/Pd/ZIF-8, the $H_{ad}$ adsorption energy is $-0.524$ eV for high $H_{ad}$ coverage ($\theta = 1.00$) and $-0.819$ eV for low $H_{ad}$ coverage ($\theta = 0.03$). In comparison, for Au/Pd, the $H_{ad}$ adsorption energy is $-0.387$ eV for $\theta = 1.00$ and $-0.517$ eV for $\theta = 0.03$ (Supplementary Fig. 10). The higher $H_{ad}$ adsorption energy indicates that hydrogen atoms are adsorbed more easily on the Pd surface, resulting in a higher concentration of hydrogen atoms on the Pd surface in Au/Pd/ZIF-8 compared to Au/Pd under varying $H_2$ conditions[36-40]. Furthermore, the potential barrier of hydrogen atoms passing through Pd (100) surface to octahedral sites is 0.64 eV for Au/Pd/ZIF-8 and 0.37 eV for Au/Pd, which illustrates that hydrogen atoms are more difficult to penetrate into Pd lattice in Au/Pd/ZIF-8 (Supplementary Fig. 11)[41]. Combining higher $H_{ad}$ adsorption energy and higher potential barrier of Au/Pd/ZIF-8, it is reasonable to conclude that Au/Pd/ZIF-8 shows higher hydrogen concentration and lower hydrogenation rate.

Upon recognizing this feature, a question naturally arises whether the higher hydrogen concentration at the Pd–ZIF-8 interface is related to electronic effects. It has been reported that electron density can influence the hydrogen concentration in Pd lattice[42]. To gain infor-mation for the electron density in Pd, we specifically perform Pd $L_3$-edge XAS to characterize Au/Pd/ZIF-8 in reference to Au/Pd. As illu-strated in Supplementary Fig. 12, the absorption coefficient ($\mu(E)$) of Au/Pd/ZIF-8 is lower than $\mu(E)$ of Au/Pd at the $L_3$-edge of Pd, suggesting the reduced electron density of Pd in Au/Pd/ZIF-8. This indicates that

the $4d$-electrons of Pd slightly transfer to ZIF-8[43]. In addition, as depicted by the Bader analysis of Pd–ZIF-8 in Supplementary Fig. 13, ZIF-8 receives 0.3 e$^-$ from Pd[44]. Given that the low electron density can promote the dissociation of $H_2$ into $H_{ad}$ on Pd surface[45,46], the addition of MOF may enhance the $H_{ad}$ concentration at Pd–ZIF-8 interface under 4% $H_2$ condition. However, we cannot make this conclusion without characterization of the interface–ZIF-8 regime, as the dynamic processes of $H_{ad}$ atoms where $H_{ad}$ atoms can spillover between the Pd–ZIF-8 interface will possibly alter the $H_{ad}$ concentration at Pd–ZIF-8 interface. In fact, in the Au/Pd/ZIF-8 structure, we can still observe the emerging $^2H$ NMR peak shift at higher field (Supplementary Fig. 14) and the XRD peak split of ZIF-8 (Supplementary Fig. 15), suggesting the underlying dynamic processes in the interface–ZIF-8 regime.

## In situ SERS characterization

To further examine the interface–ZIF-8 regime, we design the second model of Pd/ZIF-8/Au with a controllable sandwich structure for in situ SERS characterization (Fig. 3a). In this structure, Au nanoparticles are separated from Pd by ZIF-8. $H_2$ molecules are adsorbed and dis-sociated on Pd surface (i.e., Pd–ZIF-8 interface) to generate active $H_{ad}$ atoms. If the $H_{ad}$ atoms can penetrate ZIF-8 layer through hydrogen spillover and reach surface of Au nanoparticles, they will hydrogenate a probe molecule of para-nitrothiophenol (pNTP) to para-aminothiophenol (pATP)[16]. As such, the thickness of ZIF-8, namely the distance from Pd to Au where pNTP hydrogenation takes place, represents the penetration depth of hydrogen spillover from Pd–ZIF-8 interface into ZIF-8. With controllable ZIF-8 thickness, the penetration depth of hydrogen spillover can be monitored by the hydrogenation of pNTP using in situ SERS, providing information for the interface–MOF regime. In the modular construction, a 5 nm Pd layer is deposited on a silicon wafer by electron beam evaporation, and then a ZIF-8 layer with pore size of 3 nm (Supplementary Fig. 16) is prepared by chemical vapor deposition using ZnO as precursor[47]. The thickness of the ZIF-8 layer can be tailored by adjusting the amount of ZnO precursor. In detail, ZIF-8 with thicknesses of 20, 40, 60, and 120 nm is coated onto the surface of 5 nm Pd layer (Supplementary Figs. 17-20), respectively. Au nanoparticles (ca. 45 nm) with pNTP (Supplementary Figs. 21,22) are then placed on top of ZIF-8 to form Pd/ZIF-8/Au composite structures (denoted as Pd/ZIF-8-x nm/Au, in which x nm represents the thickness of ZIF-8 layer, Supplementary Fig. 23).

The hydrogenation of pNTP on Pd/ZIF-8/Au structures is per-formed using 100% $H_2$ at 25 °C and monitored by Raman spectroscopy whose signals are significantly enhanced by surface plasmon of Au (Supplementary Fig. 24). As shown in Fig. 3b, the Raman peak at 1337 cm$^{-1}$ attributed to the symmetric nitro stretching vibration of pNTP decreases during reaction duration using Pd/ZIF-8-20 nm/Au, suggesting the hydrogenation of pNTP. This reveals that the $H_{ad}$, generated from $H_2$ dissociation at Pd–ZIF-8 interface, can travel through ZIF-8 and hydrogenate pNTP on the surface of Au NPs. Notably, with the increase in ZIF-8 thickness, the hydrogenation rate of pNTP decreases as indicated by the evolution of Raman peak (Fig. 3c, Supplementary Fig. 25a, Supplementary Table 2). As the thickness of ZIF-8 is increased to 120 nm, a negligible change is observed in the Raman peak of pNTP (Fig. 3c), suggesting that $H_{ad}$ can hardly travel through ZIF-8 with such a thickness to reach Au and participate in pNTP hydrogenation. In addition, the hydrogen spillover is not able to migrate through Pd–$SiO_2$–Au interface (Supplementary Fig. 25b). The SERS characterization manifests that hydrogen spillover can indeed take place from Pd–ZIF-8 interface into ZIF-8 but with a penetration depth limit (Fig. 3d).

## AIMD theoretical calculations

Since the $H_{ad}$ atoms on Pd can travel into ZIF-8, the concentration of $H_{ad}$ atoms at the Pd–ZIF-8 will be reduced in Au/Pd/ZIF-8. In this case, the increase in surface hydrogen concentration, which has been

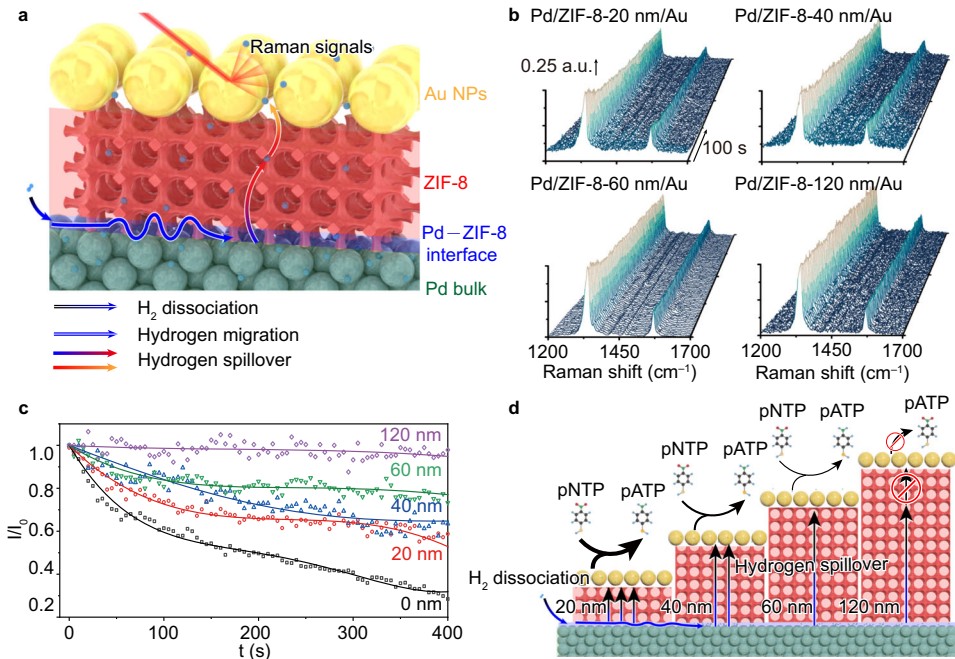

**Fig. 3 | In situ SERS characterization of hydrogen spillover in interface−ZIF-8 regime. a** Schematic illustration of Pd/ZIF-8/Au model for in situ SERS characterization[25]. The color of arrows represents the start point and the destination of hydrogen migration. **b** In situ SERS spectra of pNTP hydrogenation on Pd/ZIF-8-20, Pd/ZIF-8-40, Pd/ZIF-8-60, and Pd/ZIF-8-120 nm/Au. **c** Time-dependent Raman peak intensity for the nitro group of pNTP on Pd/ZIF-8/Au with different

thicknesses of ZIF-8 layer. **d** Schematic illustrating that hydrogen atoms can migrate from Pd through ZIF-8 to Au and participate in pNTP reduction[25]. Pd atoms: green balls; hydrogen atoms: blue balls; Au atoms: yellow balls; ZIF-8 coating: pink porous structure. The color of arrows represents the start point and the destination of hydrogen migration.

observed from quick-EXAFS at 4% $H_2$, should point to an underlying process in which $H_{ad}$ atoms can reversely spillover from ZIF-8 to Pd once the adverse balance of $H_{ad}$ concentration appears between ZIF-8 and Pd−ZIF-8 interface especially under 4% $H_2$ conditions. To clarify how the reversible $H_{ad}$ spillover occurs between Pd and ZIF-8, a model of Zn bridge is conjectured to illustrate the mechanism of hydrogen spillover. To this end, we employ AIMD simulations[48,49], combined with an explicit Pd/ZIF-8 model, to study the elementary steps in hydrogen spillover at the interface of Pd and ZIF-8 (Supplementary Figs. 26,27). To evaluate the potential barrier, constrained AIMD within a "slow-growth" method is performed to sample free energy profile[50]. The calculated potential barrier of elementary steps shows that the potential barrier of hydrogen atoms passing through Pd (100) surface to octahedral sites is 0.64 eV in the first layer (Supplementary Fig. 11), which will increase with the penetration into Pd lattice[41]. In comparison, the potential barrier of hydrogen spillover from Pd (100) surface to ZIF-8 is 1.0 eV as such a spillover has to overcome a chemisorption energy barrier[31] (Fig. 4a). Considering the increasing energy barrier of deep Pd hydrogenation, a certain portion of hydrogen atoms will spillover from Pd surface to ZIF-8 as demonstrated by the simulation (Supplementary Movie 1, Supplementary Fig. 28). In the meantime, we notice that the $H_{ad}$ in ZIF-8 has a very low barrier (0.1 eV) to migrate back to the Pd surface, which is almost a physisorption desorption[23,26] (Fig. 4b). As a result, the reverse spillover from ZIF-8 to Pd surface can take place (Supplementary Movie 2, Supplementary Fig. 29), once the balance of $H_{ad}$ atom concentration around the Pd−ZIF-8 interface is altered by dynamic processes. In this composite structure, all the dynamic processes, which involve the $H_2$ dissociation on Pd and the $H_{ad}$ spillover between Pd and ZIF-8, can be reversible in principle (Fig. 4c). We recognize that the $Zn^{2+}$ site acts as a bridge for $H_{ad}$ migration between Pd and ZIF-8. Based on the reversible hydrogen spillover at Pd−ZIF-8 interface, intuitively ZIF-8 may serve as a reservoir for hydrogen atoms−storing and releasing $H_{ad}$ atoms on-demand for their local concentration changes.

## Alkyne semihydrogenation performance of Au/Pd and Au/Pd/ZIF-8

Based on the reversible hydrogen spillover at the Pd−ZIF-8 interface, ZIF-8 may serve as a reservoir for hydrogen atoms−storing and releasing $H_{ad}$ atoms on-demand for their local concentration changes. Given the excellent activity of Pd in catalytic organic hydrogenation[44,51,52], we demonstrate selective hydrogenation of unsaturated compounds as a proof of concept. As illustrated in Fig. 5, we employ the hydrogenation of diethyl acetylenedicarboxylate (DAC), dimethyl acetylenedicarboxylate (DMAD), 2-methyl-3-butyn-2-ol (MBY), and acetylene as model reactions to assess the $H_{ad}$ concentration on the Pd surface. Two samples, Au/Pd and Au/Pd/ZIF-8, are employed to confirm the impact of hydrogen spillover on alkyne hydrogenation under light irradiation or heating conditions. The samples, Au/Pd and Au/Pd/ZIF-8, are the same as the models used for the XAFS study. Au NRs can harvest light in visible and near-infrared regions through surface plasmon to generate hot electrons and induce photothermal conversion, which would in turn promote the dissociation of $H_2$ molecules. The unique optical features of Au NRs make Au/Pd and Au/Pd/ZIF-8 excellent light-driven catalysts.

Figure 5b, c shows the hydrogenation of DAC, DMAD, MBY, and acetylene catalyzed by Au/Pd and Au/Pd/ZIF-8 under light irradiation or thermal conditions at 4% $H_2$. The light intensity has been optimized to 100 mW/cm² as shown in Supplementary Fig. 30. The catalyst maintains the morphology and structure of Au NRs@Pd@ZIF8 after reaction (Supplementary Fig. 31). The turnover frequencies (TOFs) of Au/Pd/ZIF-8 are dramatically higher than those of bare Au/Pd under both light irradiation and thermal conditions. The rate-limiting step of alkyne hydrogenation is usually assumed to be the first H-addition step[53]. As such, the $H_{ad}$ concentration is a key factor for alkyne hydrogenation. The higher $H_{ad}$ concentration will accelerate the rate of alkyne hydrogenation reaction which leads to higher alkyne TOF (Fig. 5d). In this case, the anomalously high efficiency/selectivity performance at 4% hydrogen concentration illustrates that Au/Pd/ZIF-8 has high $H_{ad}$ concentration. This result is attributed to the reversible

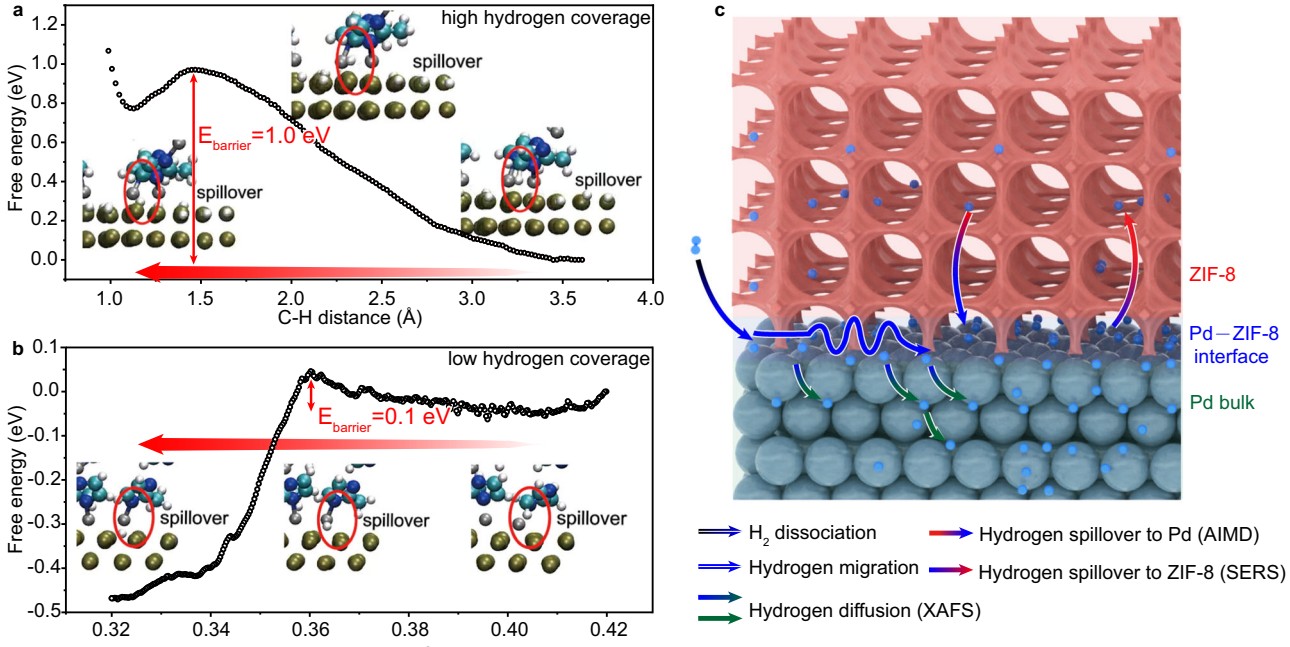

**Fig. 4 | Theoretical calculations for reversible hydrogen spillover in Pd/ZIF-8 structures.** Free energy profiles of hydrogen atom spillover **a** from Pd to ZIF-8 (the X-axis represents the distance of C–H, high hydrogen coverage) and **b** from ZIF-8 back to Pd (the X-axis represents the Z-coordinate of the hydrogen atom, low hydrogen coverage). The position of hydrogenation is illustrated by the

schematics. **c** Schematic demonstrating the methods used to examine hydrogen spillover. Pd atoms: green balls; hydrogen atoms: blue balls; ZIF-8 coating: pink porous structure. The color of arrows represents the start point and the destination of hydrogen migration.

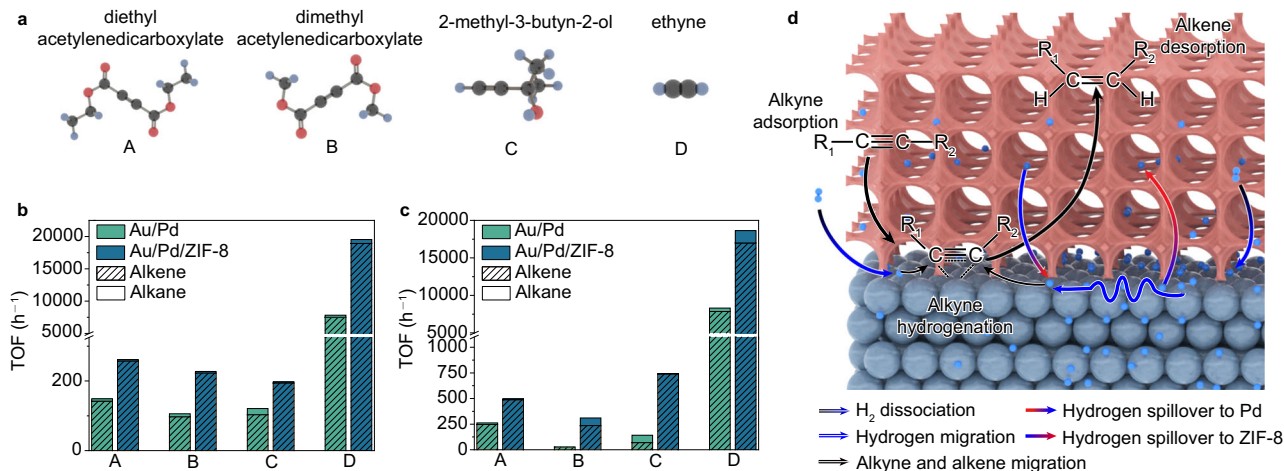

**Fig. 5 | Catalytic performance of Au/Pd and Au/Pd/ZIF-8. a** Structures of four acetylenes−diethyl acetylenedicarboxylate (A), dimethyl acetylenedicarboxylate (B), 2-methyl-3-butyn-2-ol (C), and acetylene (D). Carbon atoms: black balls; hydrogen atoms: blue balls; oxygen atoms: red balls. TOF of acetylenes hydrogenation of Au/Pd and Au/Pd/ZIF-8 under irradiation (**b**) and heating (**c**) conditions.

**d** Mechanism for catalytic hydrogenation of unsaturated hydrocarbons. Pd atoms: dark blue balls; hydrogen atoms: light blue balls; ZIF-8 coating: pink porous structure. The color of arrows represents the start point and the destination of hydrogen migration. The black arrows represent the migration alkyne and alkene molecular.

hydrogen spillover enabled by ZIF-8, which can enhance the $H_{ad}$ atom concentration (as proven by quick-EXAFS and $H_{ad}$ adsorption energy calculation) for hydrogenation reactions regardless of the catalytic method.

To summarize, we have achieved the spectroscopic visualization of reversible hydrogen spillover in metal/MOF composite structures, leveraging modular signal amplification. Our quick-XAFS characterization has identified the insensitive property of MOF coating for $H_2$ concentration. Based on $H_{ad}$ adsorption energy calculation, the $H_{ad}$ concentration of Au/Pd/ZIF-8 is higher than Au/Pd particularly at low

$H_2$ concentrations. Tracing hydrogen atoms, from Pd−MOF interface to MOF via in situ SERS, reveals that $H_{ad}$ atoms can indeed travel in MOF within a length limit. The $H_{ad}$ atoms will be released out of the MOF as confirmed by AIMD calculation once the balance of $H_{ad}$ concentration between MOF and Pd bulk is altered, which increases the $H_{ad}$ concentration on Pd surface. Enabled by this reversible process, the MOF actually works as a reservoir for effectively controlling the hydrogen concentration on Pd surface. With dynamic replenishment of surface hydrogen atoms via such an approach, Pd/ZIF-8 catalyst shows unique hydrogenation activity and selectivity for various

unsaturated compounds. This work reveals the existence and pathway of hydrogen spillover in the metal/MOF system with direct evidence, and demonstrates the potential of modular signal amplification in investigating interfacial species migration.

# Methods

## Materials

Hydrochloric acid (36–38%, AR), acetonitrile ($CH_3CN$, AR), chloroauric acid hydrated ($HAuCl_4 \cdot 4H_2O$, AR), cetyltrimethylammonium bromide (CTAB, 99.0%), ascorbic acid (AA, 99.7%), zinc nitrate hexahydrate ($Zn(NO_3)_2 \cdot 6H_2O$, 99.0%), methylbenzene (AR), and N,N-dimethylformamide (DMF, AR) were purchased from Sinopharm Chemical Reagent Co., Ltd. 2-Methylimidazole (98.0%), DAC (98%) was purchased from Energy Chemical. Benzylimidazole (98%) and DMAD (96%), 4-nitrothiophenol (pNTP, >95%) was purchased from Tokyo Chemical Industry. Potassium palladium (II) chloride ($K_2PdCl_4$), poly(vinyl pyrrolidone), and sodium borohydride were purchased from Aladdin Industrial Inc. Silver nitrate ($AgNO_3$, 99.0%) was purchased from Sigma-Aldrich. Sodium citrate ($C_6H_5Na_3O_7$, 99.0%) was purchased from J&K Scientific. MBY (98.0%) was purchased from Macklin. The water used in all experiments was deionized (DI). All chemicals were used as received without further purification.

## Synthesis of Au/Pd core–shell nanostructures

Au/Pd core–shell nanostructures were prepared according to Huang's work[54], a suspension of Au nanorods (0.5 mL, 1 mg mL$^{-1}$) was mixed with the aqueous solution of CTAB, (8.85 mL, 0.1 M), followed by the addition of aqueous $K_2PdCl_4$ (500 μL, 0.01 M) and HCl (100 μL, 1 M) solutions. Subsequently, an aqueous solution of AA (200 μL, 0.1 M) was added into the mixture, which was then shaken and kept in an oil bath at 50 °C for 2 h. The color of the mixture turned from brown to black. Finally, the products were centrifuged three times at 7200 g for 10 min. The Pd/Au molar ratio was determined to be 1.62.

## Synthesis of Au/Pd/ZIF-8 composite structures

In a typical synthetic procedure, a suspension of Au/Pd (0.05 mL, 1 mg mL$^{-1}$ for Au) was mixed with a 10-mL methanol solution of 2-methylimidazole (10.25 mg) and a 10-mL methanol solution of $Zn(NO_3)_2 \cdot 6H_2O$ (37.2 mg), which was then kept undisturbed at 0 °C for 10 min. The products were centrifuged three times at 7200 g for 10 min. The precipitation was redispersed in $CH_3CN$.

## Quick-EXAFS measurements

The quick-EXAFS measurements were performed at the beamline 44 A in Taiwan Photon Source (TPS). TPS 44 A was constructed with a quick-scanning monochromator (Q-Mono) which could select the photon energy and do quick scans. The acquired time for a full quick-scanning EXAFS spectrum was less than 100 ms over 1000 eV. The beam size was 60 (H) × 200 (V) μm$^2$. The samples were characterized in the home-made EXAFS cell with 20 sccm $H_2$ or 4% $H_2$ and 96% $N_2$. EXAFS data processing was performed using the Larch software[55]. In order to fit thousands of EXAFS data, we expanded the program with a batch processing script. Every five data were merged before processing so that the real time resolution is 0.5 s. We fitted the first coordination shell with a single Pd–Pd path. The fitting k range is 3.0~11.1 Å$^{-1}$ and the R range is 1.8~2.9 Å. The standard error of fitting results of R is below 1%. No data were excluded from the analyses.

## Synthesis of Pd/ZnO composite structures

In a typical synthetic procedure, a silicon wafer was covered with a 5 nm Pd film by electron beam evaporation using K.J. Lesker, LAB 18 E-Beam Evaporator. ZnO film with different thickness was deposited at 240 °C by 500 ms pulse times and 2000 ms purge times of diethyl zinc/oxygen plasma precursor pulses (Picosun, Sunale R-200

Advanced). The thicknesses of ZnO layer were controlled by the cycles of deposition.

## Synthesis of Pd/ZIF-8 composite structures

After deposition of different thicknesses of ZnO layer, the ZnO-coated substrates were placed in a closed reactor vessel with powder of 2-methylimidazole and the spherical valve was turned on toward the monoblock pump to vacuumize vessel. After pumping vacuum, the vessel was placed in the oven and heated to 140 °C. 2 h were needed for ZIF-8 layer synthesis with different thicknesses of ZnO precursor.

## Preparation of Au nanoparticles film

A total of 55 nm Au nanoparticles were prepared by Frens' method[56]. In total, 1.5 mL (1 wt%) sodium citrate aqueous solution was added into 200 mL DI water, and the solution was heated until boiling. After that, 2.4 mL (1 wt%) chloroauric acid hydrated was added into the boiling solution. 55 nm Au nanoparticles were obtained under boiling condition after 40 min. The monolayer film of Au nanoparticles was self-assembled according to a two-phase protocol. First, 10 mL Au nanoparticles sols were added into a petri dish (diameter 125 mm). Subsequently, 2-3 mL methylbenzene was added to Au sols until a complete methylbenzene layer was assembled on the water layer. Then 1 mL N,N-DMF was added into the mixed solution, and a number of monolayer films appeared. The petri dish was placed in a fume cupboard overnight to make the methylbenzene evaporate. The films were transferred to substrates for further experiments.

## Synthesis of Pd/ZIF-8/Au composite structures

The substrate was Pd/ZIF-8 composite structure mentioned above. 1 mL 0.1 M pNTP was immersed in the petri dish which contained Au nanoparticles film (diameter 125 mm) and was let stand for 1 h. The molecules of pNTP were adsorbed on the surface of Au nanoparticles completely. Then, the Pd/ZIF-8 substrate was dipped into Au nanoparticles films at a small angle and pulled out gently, which formed Pd/ZIF-8/Au composite structures.

## In situ SERS measurements

The Raman spectra were recorded on a WITec alpha 300 R confocal Raman microscope. The excitation source wavelength was 633 nm, and the diffraction grating was 600 grooves/mm. A 50× microscope objective which had a 0.55 numerical aperture was used. The laser power was controlled at 1 mW. All experiments were performed in the special Raman cell with atmosphere control. The silicon wafer with Pd/ZIF-8/Au structure was located in a quartz cell which can control atmosphere by three-way valve. We turned the three-way valve toward monoblock pump to vacuumize the cell and then turned valve to pure hydrogen balloon to change the air atmosphere to 100% hydrogen atmosphere. The in situ SERS measurements were performed at 25 °C. We collected the signals after the environment change to hydrogen atmosphere. The integral time that we used was 5 s and we collected the signals for 400 s of each sample.

## Catalytic hydrogenation reactions

The hydrogenation of DAC was carried out in an 18 cm tall quartz tube. The Au/Pd/ZIF-8, Au/Pd or Pd/ZIF-8 catalysts containing 0.1 mg Pd were added into the tube, followed by the addition of 10 μL DAC. $CH_3CN$ was added into the solution until the total volume of the mixture reached 1 mL. The hydrogenation procedures for MBY and DMAD were similar to that for DAC by the addition of 10 μL substrate. A balloon filled with 4% $H_2$ was used to enclose the system and provide hydrogen source. After exhausting air with $H_2$, the catalytic reactions were carried out at room temperature (25 °C) under light irradiation. Light irradiation was carried out using a Xe lamp (Solaredge 700, China) at 100 mW cm$^{-2}$. The heating condition was performed in constant temperature oil-bath pans with 50 °C. After the reaction, the

catalysts were removed by filtration, and the resulting solution was analyzed by a gas chromatography–mass spectrometry (GC–MS, 7890A and 5975C, Agilent). The hydrogenation of acetylene was conducted in a home-made flow reactor. The Au/Pd/ZIF-8 and Au/Pd catalysts containing 0.1 mg Pd were dispersed on the glass fiber filter membrane. In total, 100 sccm 1% $C_2H_2$ and 4% $H_2$, balanced with Ar, were blown into the reactor to remove the air for 10 min, and then the flow rate was lowered to 40 sccm for reaction. Light irradiation was carried out using a Xe lamp (Solaredge 700, China) at 100 mW cm$^{-2}$ at 25 °C. The heating condition was controlled by heater band at 50 °C. The gaseous products were analyzed by a gas chromatography (GC, GC2014 + AFSC, 230C, Shimadzu).

## Data availability
The data used to support the study and to generate figures and tables are available from the corresponding author upon request. Source Data are included with this manuscript. Figshare https://doi.org/10.6084/m9.figshare.23912715. Source data are provided with this paper.

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

## Acknowledgements

This work was supported by National Key R&D Program of China (2020YFA0406103), NSFC (22122506, 22232003, 22075267, U23A2091, 51902311, 22150610467), Strategic Priority Research Program of CAS (XDB0450401), Youth Innovation Promotion Association of CAS (2019444), West Light Foundation of Chinese Academy of Sciences (xbzg-zdsys-202209), and Fundamental Research Funds for the Central Universities (20720220007, WK2060000039, KY2140000031). Quick-scanning X-ray absorption spectroscopy was performed on the beamline (TPS 44A) of the National Synchrotron Radiation Center. XAS measurements were performed at beamline 4B7A of the Beijing Synchrotron Radiation Facility (BSRF) and beamline BL14W1 of the Shanghai Synchrotron Radiation Facility (SSRF). XRD measurements were performed at beamline 4B9A of BSRF and BL14B of SSRF. We thank the USTC Center for Micro- and Nanoscale Research and Fabrication for their support. The numerical calculations in this work were performed on the supercomputing system in the Supercomputing Center of the University of Science and Technology of China. This work was partially carried out at the Instruments Center for Physical Science, University of Science and Technology of China. This work was partially carried out at the Instruments Center for Physical Science, University of Science and Technology of China. We thank Chih-Wen Pao for assistance in collecting the Quick-scanning X-ray absorption spectroscopy signals.

## Author contributions

Y.X. and R.L. conceived the research and designed the experiments. Q.L., H.H., Y.L., D.D., W.Z., Y.J. and K.W. performed the synthesis and characterization of samples. Q.L, D.D., and D.C. performed the XAFS studies. W.X., G.Z., and S.C. analyzed the XAFS data. H.S. calculated the AIMD. Q.L., J.L., Y.D., W.G., R.L., L.S. and Y.X. analyzed the data and wrote the manuscript. All authors discussed the results and commented on the manuscript. Q.L., W.X., H.H. and H.S. equally contributed to this work.

## Competing interests

The authors declare no competing interests.
