## [Peer Review File · Nature Communications]

Spectroscopic visualization of reversible hydrogen spillover between palladium and metal-organic frameworks toward catalytic semihydrogenationReviewers' comments:

Reviewer #1 (Remarks to the Author):

In this communication, the authors report their recent research effort on investigation of hydrogen spillover phenomena in the Pd/ZIF-8 system using a number of dynamic spectroscopic methods (quick-EXAFS & in-situ SERS). Some of findings were further supported by ab initio molecular dynamics (AIMD) calculation. Overall speaking, this work provides readers a more detailed view on this important pair of material combination, which is Pd, Pd-ZIF-8 interface, and ZIF-8, under hydrogen spillover or common hydrogenation conditions. Therefore, the work can be recommended for its publication in this Journal after following revisions.

1. It has been well-known that the H(ad) from hydrogen spillover can pass through ZIF-8 with a limited travel distance (for example, Ref 25). Therefore, the statements of "... it remains unclear whether they may migrate into MOF or back to Pd-MOF interface (lines 83 to 84) ..." and "... If the Had atoms can penetrate ZIF-8 layer through hydrogen spillover and reach surface of Au nanoparticles ... (lines 209 to 210)" may have be rephrased accordingly.
2. Figure 3(a & d): This sandwich configuration (Pd/ZIF-8/Au) for H(ad) diffusion path length determination has been similarly investigated in Pt/ZIF-8/ZIF-67 system. Thus Ref. 25 should also be cited here.
3. Preparation of Pd/ZIF-8/Au composite structures is not very clear. In fact, a better description on "Method" should be targeted in revision.
4. Figure S13(b): This image (Pd/ZIF-8-40 nm) does not indicate that the ZIF-8 coating is around 40 nm. More convincing data should be supplemented.
5. Comments on used samples: Any observable changes in terms of morphological change after reactions or measurements?
6. Au phase should be denoted in Figure 2a and Figure 3a, although it is deducible from these figures.
7. Lines 157 to 161: The percentage expansion from metallic Pd phase to PdH(beta) can be stated here in order to give readers a more quantitative understanding.

Reviewer #2 (Remarks to the Author):

The authors proposed the reversible hydrogen spillover between Pd and MOF through their interface. Several characterizations including quick-EXAFS and in situ SERS were applied to demonstrate the

proposed spillover mechanism. However, some results and discussion are inconsistent, and several questions need to be further addressed. The current version of the manuscript does not meet the high standard of nature communications. Below are the detailed comments:

(1) For the Quick-EXAFS experiments, the result in Fig. 2d shows that k_R for Au/Pd/ZIF-8 under 4% H₂ is 0.9×10^{-4} , while the main text (Page 9, line 166) indicates that k_R for Au/Pd/ZIF-8 under 4% H₂ is 1.3×10^{-4} . This contradiction makes the following discussion (line 162-195) confused. The author indicated that "As such, the results above imply that under 4% H₂ condition, Au/Pd/ZIF-8 has a higher hydrogen concentration on Pd surface (i.e., Pd-ZIF-8 interface in this case) than normal (i.e., the case without ZIF-8 coating)", do the authors mean that the hydrogenation rate over Au/Pd/ZIF-8 was higher under 4% H₂? However, the result in Fig. 2d showed that k_R for Au/Pd/ZIF-8 was lower than Au/Pd under 100% H₂ or 4% H₂, which means hydrogenation rate for Au/Pd/ZIF-8 was lower than Au/Pd. The authors only try to elaborate the result under 4% H₂, what makes the difference under 100% H₂?

(2) The Quick-EXAFS results indicated that the formation rate of PdH was decreased by coating ZIF-8, while Hiroshi Kitagawa et al. indicated that Pd covered with MOF can enhanced the formation rate of PdH (Nat. Mater. 13, 802-806 (2014), which was also cited by the authors). If the proposed spillover behavior exists, what make the difference?

(3) For the in situ SERS experiment, the authors ascribed the activity difference to the spillover distance, however this comparative experiment is not rigorous. How to confirm each silicon wafer contains the same active site of Pd and Au? The author did not demonstrate that the Pd layer was fully covered with ZIF-8 layer. Actually, during the formation of ZIF-8 layer, large pores could also generate, with which the Pd layer could directly interact with pNTP and results in higher activity. The increased thickness of ZIF-8 could decrease the defect pores and thus decrease the activity.

(4) Concerning the catalytic performance in Fig. 5, ZIF-8 could serve as a cage to enrich reactant concentration, which could also enhance the activity. How to exclude reactant enrichment effect?

(5) In page 24 line 532, the authors indicated that catalysts containing 0.1 mg Pd were added into the tube. How to accurate keep the catalysts containing the same weight of Pd? The weight of samples should change with coating ZIF-8, while the authors did not report the metal loading of Pd for Pd/ZIF-8.

Reviewer #3 (Remarks to the Author):

Liu et.al. presents the experimental and computational observation of hydrogen spillover between palladium and metal-organic framework. In this work, the authors demonstrate that hydrogen spillover can be indirectly observed using markers for hydrogen dissolution in the Pd lattice (i.e. increase in Pd-Pd distance) and hydrogen diffusion across the MOF layer (i.e. hydrogenation of a compound deposited on the other side of the MOF). The HSPE is confirmed by the increase rate of organic hydrogenation

reactions on the prepared Pd/MOF material over Pd. A critical inconsistency between reporting of data in the Fig. 2 and in text needs to be addressed before, as it might affect the reliability of the entire manuscript. The misunderstanding and misuse of the terminology of H spillover appear throughout the entire manuscript. There are too many speculations in the mechanism deduction. I can understand the complication of this system. Also H is not easy to be detected, particularly to distinguish H₂, atomic H, proton or hydride. Thus, I think detailed theoretical studies are necessary to make the conclusions more robust. Plus, the authors might check if different H pressure will affect their H spillover mechanism. For example, defining the active sites on ZIF and Pd or Pd/ZIF interface shall be necessary. In addition, subtle improvements for the terminologies, Figure presentation and mechanistic discussions are necessary.

1. For the entirety of the manuscript, the use of the term “hydrogen spillover” might lead to confusion. Conventionally, hydrogen spillover only refers to the migration of hydrogen from one phase (e.g. metal) to another (the MOF, in the case of this study for example).

(1) In Page 4, Fig 1a, the H diffusion on Pd surface (a single phase) shall not be called H spillover.

(2) In Page 4, In Fig 1b, usually we can say H spillover is a process from H₂ dissociation on ZIF8, followed by atomic H migrating from ZIF8, via Pd-ZIF9 interface to Pd bulk. H hopping/migration on Pd-ZIF 8 interface shall not call H spillover. Else, H spillover from ZIF8 to Pd bulk.

(3) In page 5 line 74: “which can then diffuse into bulk lattice of Pd in addition to spillover on surface”

(4) In page 5 line 81: “When Pd is interfaced with MOF (Fig. 1b), hydrogen spillover

(5) 82 pathways would become elusive.” Does the authors mean that the H diffusion on the Pd-MOF interface would become elusive?

2. In page 8, Fig 2: How do the authors distinguish H alpha with H beta?

3. In page 8 line 155: “which gradually evolves from PdH_α to PdH_β with increase of H_α concentration”. Please can the author elaborate this sentence. Can you explain your definition of H alpha and H beta.

4. In page 9 line 166: “As the atmosphere is switched to 4% H₂, k_R is lowered to 1.3×10⁻⁴ Å s⁻¹ for Au/Pd/ZIF-8 and 0.9×10⁻⁴ Å s⁻¹ for Au/Pd.”

Furthermore, starting at line 176 “under 4% H₂ condition, Au/Pd/ZIF-8 has a higher hydrogen concentration on Pd surface (i.e., Pd-ZIF-8 interface in this case) than normal (i.e., the case without ZIF-8 coating).”

However, the cited data are not consistent with the ones presented in Figure 2d and e, where it shows higher rate of Pd-Pd distance change on Au/Pd than that for Au/Pd/ZIF-8. If the Figure is correct, then all the conclusions at the end of page 9 are wrong; the Au/Pd/ZIF-8 does not exhibit high hydrogenation rate at low H₂ concentration. This is crucial seems all of the following discussions hinge on the conclusions derived from the EXAFS analysis. On the other hand, if the Figure is wrong, then the Kr_{100%}/Kr_{4%} ratios reported in Figure 2e and cited in page 9 lines 168-169 are also wrong.

5. In page 10 line 185: “suggesting the reduced electron density of Pd in Au/Pd/ZIF-8. This indicates that the 4d-electrons of Pd slightly transfer to ZIF-8” This shall be easy and straightforward analysis by DFT. It

would be great if the author can prove this charge transfer by DFT. Also, The ZIF is also complicated. Will this e of Pd go to which part of ZIF?

6. In page 10 lines 189-195, “the dynamic process of Had atoms” is confusing. Had atoms itself is not a process. Can the specify here what type of dynamic process occurring and why is it “dynamic”?

7. In Figure 4, the resolution of the images is too low. In connection to this, in Figure 4b, where the H supposed goes from ZIF down to Pd, it is confusing why the x-axis of the plot indicates that the hydrogen atom is moving up (z coordinate is increasing). In connection to this, the following discussions in page 14 are confusing:

a. Line 260: “In comparison, the potential barrier of hydrogen spillover from Pd (100) surface to ZIF-8 is 1.0 eV (Fig. 4a)”. However, looking at Figure 4a, the barrier of migration from Pd to ZIF seems to be only 0.2 eV (calculating the difference between the peak and the first local minima: $1.0 - 0.8 = 0.2$ eV)

b. Line 264: “In the meantime, we notice that the Had in ZIF-8 has a very low barrier to migrate back to the Pd surface (Fig. 4b).” Please cite the value of the barrier for the reverse reaction. Also can the authors comment on why the forward and reverse migration of H spillover between Pd and ZIF have different barriers? Do they proceed via different mechanisms?

8. With regards to the hydrogenation of organic compounds discussed in pages 15-16, can the authors elaborate on the envisioned mechanism of such reactions? The HSPE is provided as the reason for the improved activity of Pd/ZIF-8 compared to Pd. However, the interface between Pd and ZIF-8 seems to be inaccessible to reactants or adsorbates with significant size such as the organic compounds used. Thus, it is not clear how organic compounds would bind to the Pd/ZIF-8 catalyst and how hydrogen spillover participates in the reaction. In addition, if a mechanism similar to that depicted in Fig. 3d is assumed, should not the requirement of Had diffusion across the ZIF-8 layer, at least in principle, slow down the hydrogenation of organic compounds compared to the bare Pd?

Reviewer #1 (Remarks to the Author):

In this communication, the authors report their recent research effort on investigation of hydrogen spillover phenomena in the Pd/ZIF-8 system using a number of dynamic spectroscopic methods (quick-EXAFS & in-situ SERS). Some of findings were further supported by ab initio molecular dynamics (AIMD) calculation. Overall speaking, this work provides readers a more detailed view on this important pair of material combination, which is Pd, Pd-ZIF-8 interface, and ZIF-8, under hydrogen spillover or common hydrogenation conditions. Therefore, the work can be recommended for its publication in this Journal after following revisions.

We really appreciate the referee's highly positive evaluation of our work, and are grateful to the referee for his/her comments and suggestions to help us further improve the quality of our manuscript. We have carefully revised the manuscript and sincerely hope that our revisions have satisfactorily addressed the referee's concerns.

1. It has been well-known that the H(ad) from hydrogen spillover can pass through ZIF-8 with a limited travel distance (for example, Ref 25). Therefore, the statements of "... it remains unclear whether they may migrate into MOF or back to Pd-MOF interface (lines 83 to 84) ..." and "... If the H_{ad} atoms can penetrate ZIF-8 layer through hydrogen spillover and reach surface of Au nanoparticles ... (lines 209 to 210)" may have be rephrased accordingly.

We thank the referee for his/her thoughtful comments. The hydrogen spillover of the Pt-MOF interface has been well investigated experimentally (*Nat. Commun.* **9**, 3778 (2018)). However, the hydrogen spillover of the Pd-MOF interface was only studied theoretically (*Int. J. Hydrogen Energy* **37**, 5081-5089 (2012)) while it lacks convincing experimental evidence. Considering the stronger interaction between Pd and H, it is difficult to directly draw similar conclusions to the Pt-MOF interface in the Pd-MOF system. For this reason, we have designed this series of experiments to verify whether hydrogen may migrate into MOF or back to Pd-MOF interface. According to the suggestion, we have rephrased the expression in the revised manuscript.

2. Figure 3(a & d): This sandwich configuration (Pd/ZIF-8/Au) for H(ad) diffusion path length determination has been similarly investigated in Pt/ZIF-8/ZIF-67 system. Thus Ref. 25 should also be cited here.

We thank the referee for bringing this to our attention. According to the suggestion, we have now cited Ref. 25 in Figure 3(a & d) in the revised manuscript.

3. Preparation of Pd/ZIF-8/Au composite structures is not very clear. In fact, a better description on "Method" should be targeted in revision.

We thank the referee for his/her valuable suggestion. We have supplemented this section in detail in the revised manuscript.

4. *Figure S13(b): This image (Pd/ZIF-8-40 nm) does not indicate that the ZIF-8 coating is around 40 nm. More convincing data should be supplemented.*

We thank the referee for his/her insightful suggestion. We have changed Figure S14(b) to another cross-sectional SEM image that can better illustrate the thickness of ZIF-8 coating.

Fig. S14 | Cross-sectional SEM images of (a) Pd/ZIF-8-20 nm, (b) Pd/ZIF-8-40 nm, (c) Pd/ZIF-8-60 nm, and (d) Pd/ZIF-8-120 nm.

5. *Comments on used samples: Any observable changes in terms of morphological change after reactions or measurements?*

We thank the referee for his/her insightful comments. According to the suggestion, we have included the result in the Supplementary Information (Supplementary Fig. 27). The TEM result shows that the morphology and structure of Au NRs@Pd@ZIF8 are well maintained after catalytic reactions.

Fig. S27 | TEM image of Au NRs@Pd@ZIF8 hybrid structures after catalytic reactions.

6. *Au phase should be denoted in Figure 2a and Figure 3a, although it is deducible from these figures.*

We thank the referee for his/her helpful suggestion. We have added the Au phase to Figure 2a

and Figure 3a according to the suggestion.

7. Lines 157 to 161: The percentage expansion from metallic Pd phase to PdH(beta) can be stated here in order to give readers a more quantitative understanding.

We thank the referee for his/her insightful suggestion. We have now included the expansion rate in the revised manuscript.

Reviewer #2 (Remarks to the Author):

The authors proposed the reversible hydrogen spillover between Pd and MOF through their interface. Several characterizations including quick-EXAFS and in situ SERS were applied to demonstrate the proposed spillover mechanism. However, some results and discussion are inconsistent, and several questions need to be further addressed. The current version of the manuscript does not meet the high standard of nature communications. Below are the detailed comments:

We are grateful to the referee for his/her comments and suggestions to help us significantly improve the quality of our manuscript. We have carefully revised the manuscript and sincerely hope that our revisions have satisfactorily addressed the referee's concerns.

(1) For the Quick-EXAFS experiments, the result in Fig. 2d shows that k_R for Au/Pd/ZIF-8 under 4% H_2 is 0.9×10^{-4} , while the main text (Page 9, line 166) indicates that k_R for Au/Pd/ZIF-8 under 4% H_2 is 1.3×10^{-4} . This contradiction makes the following discussion (line 162-195) confused. The author indicated that "As such, the results above imply that under 4% H_2 condition, Au/Pd/ZIF-8 has a higher hydrogen concentration on Pd surface (i.e., Pd-ZIF-8 interface in this case) than normal (i.e., the case without ZIF-8 coating)", do the authors mean that the hydrogenation rate over Au/Pd/ZIF-8 was higher under 4% H_2 ? However, the result in Fig. 2d showed that k_R for Au/Pd/ZIF-8 was lower than Au/Pd under 100% H_2 or 4% H_2 , which means hydrogenation rate for Au/Pd/ZIF-8 was lower than Au/Pd. The authors only try to elaborate the result under 4% H_2 , what makes the difference under 100% H_2 ?

We thank the referee for his/her insightful comments and suggestions. We have now modified our typo in the revised manuscript. k_R represents the hydrogenation rate of PdH_x , but is affected by many factors, such as surface defects and charge transfer interactions (*J. Catal.* **43**, 376-379 (1976)). For Au/Pd and Au/Pd/ZIF-8, it is complicated to interpret the difference in k_R of the two samples. However, in the same sample, k_R with different hydrogen concentrations can be directly compared. According to A. Borgschulte's research, the relationship between the palladium hydride hydrogenation rate (R) and the applied pressure (p) is established (*Phys. Rev. B* **78**, 094106 (2008)). It was acknowledged that the hydrogenation rate (R) is proportional to the surface hydrogen concentration according to the Fick's law of diffusion. The relationship between R and p can be simplified as follows:

$$\frac{R_1}{R_2} = \sqrt{\frac{p_1}{p_2}}$$

As mentioned above, $R \propto k_r, k_{CN}$ and $\frac{p_{100\%}}{p_{4\%}} = 5$. For AuPd, $\frac{k_{R100\%}}{k_{R4\%}} = 5.53$, which is in good agreement with the theory. However, $\frac{k_{R100\%}}{k_{R4\%}} = 1.91$ for AuPd@ZIF-8 is quite different from the theoretical results. The results mentioned above imply that AuPd@ZIF-8 has a higher apparent hydrogen pressure or surface hydrogen concentration under 4% H_2 conditions.

(2) The Quick-EXAFS results indicated that the formation rate of PdH was decreased by coating ZIF-8, while Hiroshi Kitagawa et al. indicated that Pd covered with MOF can enhanced the formation rate of PdH (Nat. Mater. 13, 802-806 (2014), which was also cited by the authors). If the proposed spillover behavior exists, what make the difference?

We thank the referee for his/her insightful comments and suggestions. The kinetics of the hydrogenation of Pd nanocubes and Pd@HKUST-1 were investigated by isothermal hydrogenation profiles at 303 K in the work of Hiroshi Kitagawa. They only detected the amount of hydrogen adsorbed in samples that may not form PdH_x. The results of isothermal hydrogenation profiles implied that the HKUST-1 could enhance the speed of hydrogen storage but not the formation rate of PdH_x. However, our Quick-EXAFS results can acquire the origin information of the Pd lattice which illustrates the formation rate of PdH_x. In addition, it should not be ignored that the MOF coatings used in our work are different which would strongly affect the surface chemical environment.

(3) For the in situ SERS experiment, the authors ascribed the activity difference to the spillover distance, however this comparative experiment is not rigorous. How to confirm each silicon wafer contains the same active site of Pd and Au? The author did not demonstrate that the Pd layer was fully covered with ZIF-8 layer. Actually, during the formation of ZIF-8 layer, large pores could also generate, with which the Pd layer could directly interact with pNTP and results in higher activity. The increased thickness of ZIF-8 could decrease the defect pores and thus decrease the activity.

We thank the referee for his/her thoughtful comments. The ZIF-8 layer used in our work was synthesized by Stassen's method (Nat. Mater. 15, 304-310 (2016)) which was proven to form continuous and compact coatings. To further confirm that the Pd layer was fully covered with the ZIF-8 layer, top-view SEM images have been collected (Supplementary Fig. 15). As shown in Fig. S14, no obvious large gaps were observed at the surface of the ZIF-8 layer. Moreover, AFM imaging illustrated that the surface of the ZIF-8 film was completely covered by ZIF-8 (Supplementary Fig. 16). The SEM and AFM results have demonstrated that the Pd layer was fully covered with the ZIF-8 layer.

Fig. S15 | SEM images of (a) Pd/ZIF-8-20 nm, (b) Pd/ZIF-8-40 nm, (c) Pd/ZIF-8-60 nm, and (d) Pd/ZIF-8-120 nm.

Fig. S16 | AFM analysis of ZIF-8 films with different thicknesses: (a) 20 nm, (b) 40 nm, (c) 60 nm, and (d) 120 nm.

(4) Concerning the catalytic performance in Fig. 5, ZIF-8 could serve as a cage to enrich

reactant concentration, which could also enhance the activity. How to exclude reactant enrichment effect?

We thank the referee for his/her insightful comments and suggestions. To determine whether ZIF-8 could enrich reactant concentration, 10 mg ZIF-8 which was far more than the amount used in the catalyst was added to 1 mL CH₃CN containing 10 μ L substrate (DAC, MBY, or DMAD). The solution was stirred overnight to ensure that the substrate could be adequately adsorbed onto ZIF-8. Next, we tested the amount of substrate by GC-MS and contrasted the result of solution without ZIF-8. As shown in Table R1, a small decrease (1.4%~8.9%) in solution concentration was detected with the ZIF-8 coating but the dramatically higher TOF (85.7%~134.0%) was observed. Therefore, the contribution of the reactant enrichment effect to the hydrogenation performance can be excluded.

Table R1 | Amount of substance of different substrates with ZIF-8 after 10 h adsorption.

Substrates	Amount of substance (with ZIF-8)/ μ mol	Amount of substance (without ZIF-8)/ μ mol
DAC	56.8	62.4
MBY	101.76	103.2
DMAD	78.9	81.3

(5) In page 24 line 532, the authors indicated that catalysts containing 0.1 mg Pd were added into the tube. How to accurate keep the catalysts containing the same weight of Pd? The weight of samples should change with coating ZIF-8, while the authors did not report the metal loading of Pd for Pd/ZIF-8.

We thank the referee for his/her valuable comments and suggestions. The samples used in acetylene hydrogenation were dispersed in water or CH₃CN which could be quantified by inductively coupled plasma optic emission spectrometer (ICP-OES). As shown in Table R2, the metal loading of Pd for Pd/ZIF-8 for NMR is 4.20%, and the metal loading of Pd for Pd/ZIF-8 for XRD is 5.42%.

Table R2 | Element content detected by ICP-OES.

Samples	Element content (μ g/mL)
Pd NCs	Pd: 13.779
Pd/ZIF-8 for NMR	Pd: 21.166; Zn: 73.468
Pd/ZIF-8 for XRD	Pd: 13.779; Zn: 34.495
Au/Pd	Pd: 3.016; Au: 3.515
Au/Pd/ZIF-8	Pd: 1.048; Au: 1.224; Zn: 2.605

Ten microliters of Pd NC, Au/Pd and Au/Pd/ZIF-8 solutions were dissolved in 1 mL of aqua regia and then diluted to 5 mL with water. A total of 2.52 mg Pd/ZIF-8 for NMR and 1.27 mg Pd/ZIF-8 for XRD were dissolved in 1 mL aqua regia and diluted to 5 mL with water. The

metal loading of Pd for Pd/ZIF-8 for NMR is 4.20%, and the metal loading of Pd for Pd/ZIF-8 for XRD is 5.42%.

Reviewer #3 (Remarks to the Author):

Liu et.al. presents the experimental and computational observation of hydrogen spillover between palladium and metal-organic framework. In this work, the authors demonstrate that hydrogen spillover can be indirectly observed using markers for hydrogen dissolution in the Pd lattice (i.e. increase in Pd-Pd distance) and hydrogen diffusion across the MOF layer (i.e. hydrogenation of a compound deposited on the other side of the MOF). The HSPE is confirmed by the increase rate of organic hydrogenation reactions on the prepared Pd/MOF material over Pd. A critical inconsistency between reporting of data in the Fig. 2 and in text needs to be addressed before, as it might affect the reliability of the entire manuscript. The misunderstanding and misuse of the terminology of H spillover appear throughout the entire manuscript. There are too many speculations in the mechanism deduction. I can understand the complication of this system. Also H is not easy to be detected, particularly to distinguish H₂, atomic H, proton or hydride. Thus, I think detailed theoretical studies are necessary to make the conclusions more robust. Plus, the authors might check if different H pressure will affect their H spillover mechanism. For example, defining the active sites on ZIF and Pd or Pd/ZIF interface shall be necessary. In addition, subtle improvements for the terminologies, Figure presentation and mechanistic discussions are necessary.

We are grateful to the referee for his/her comments and suggestions to help us significantly improve the quality of our manuscript. We have carefully revised the manuscript and sincerely hope that our revisions have satisfactorily addressed the referee's concerns.

1. For the entirety of the manuscript, the use of the term "hydrogen spillover" might lead to confusion. Conventionally, hydrogen spillover only refers to the migration of hydrogen from one phase (e.g. metal) to another (the MOF, in the case of this study for example).

(1) In Page 4, Fig 1a, the H diffusion on Pd surface (a single phase) shall not be called H spillover.

(2) In Page 4, In Fig 1b, usually we can say H spillover is a process from H₂ dissociation on ZIF8, followed by atomic H migrating from ZIF8, via Pd-ZIF8 interface to Pd bulk. H hopping/migration on Pd-ZIF 8 interface shall not call H spillover. Else, H spillover from ZIF8 to Pd bulk.

(3) In page 5 line 74: "which can then diffuse into bulk lattice of Pd in addition to spillover on surface"

(4) In page 5 line 81: "When Pd is interfaced with MOF (Fig. 1b), hydrogen spillover

(5) 82 pathways would become elusive." Does the authors mean that the H diffusion on the Pd-MOF interface would become elusive?

We thank the referee for his/her insightful comments and suggestions. The movement of hydrogen atoms on the Pd surface or Pd–MOF interface was defined as hydrogen migration; the movement of hydrogen atoms from the Pd phase to the ZIF-8 phase or from the ZIF-8 phase to the Pd phase was named as hydrogen spillover; and the movement of hydrogen atoms from the Pd surface to the Pd lattice forming PdH_x was defined as hydrogen diffusion. According to

the suggestion, we have made amendments in the revised manuscript. As for comment (5), “hydrogen spillover pathways would become elusive” means that hydrogen atoms could randomly migrate to the ZIF-8 coating or Pd-MOF interface. It is difficult to detect and distinguish the migration directions.

2. In page 8, Fig 2: How do the authors distinguish H alpha with H beta?

We thank the referee for his/her thoughtful comments. The hydrogen atoms would only occupy the octahedral sites regardless of PdH_α or PdH_β (*J. Am. Chem. Soc.* **138**, 10238-10243 (2016)). As such, the increase in the Pd–Pd bond distance is related to the occupation of the octahedral sites. When Pd–Pd interatomic distances reach a constant value, the hydrogen atoms occupy maximum octahedral sites. This state was commonly defined as PdH_β. The state when Pd–Pd interatomic distances increase steadily was defined as PdH_α. The turning point in the increase was defined as the mixed phase of PdH_α and PdH_β.

3. In page 8 line 155: “which gradually evolves from PdH_α to PdH_β with increase of H_{ad} concentration”. Please can the author elaborate this sentence. Can you explain your definition of H alpha and H beta.

We thank the referee for his/her insightful comments and suggestions. Similar to Comment 2, the hydrogen atoms would only occupy the octahedral sites regardless of PdH_α or PdH_β (*J. Am. Chem. Soc.* **138**, 10238-10243 (2016)). As such, the increase in the Pd–Pd bond distance can be correlated with the occupation of the octahedral sites. When Pd–Pd interatomic distances reach a constant value, the hydrogen atoms occupy maximum octahedral sites. We defined this state as PdH_β. The state when the Pd–Pd interatomic distances increase steadily was defined as PdH_α. The turning point in the increase was defined as mixed phase of PdH_α and PdH_β.

4. In page 9 line 166: “As the atmosphere is switched to 4% H₂, k_R is lowered to $1.3 \times 10^{-4} \text{ \AA s}^{-1}$ for Au/Pd/ZIF-8 and $0.9 \times 10^{-4} \text{ \AA s}^{-1}$ for Au/Pd.” Furthermore, starting at line 176 “under 4% H₂ condition, Au/Pd/ZIF-8 has a higher hydrogen concentration on Pd surface (i.e., Pd–ZIF-8 interface in this case) than normal (i.e., the case without ZIF-8 coating).”

However, the cited data are not consistent with the ones presented in Figure 2d and e, where it shows higher rate of Pd-Pd distance change on Au/Pd than that for Au/Pd/ZIF-8. If the Figure is correct, then all the conclusions at the end of page 9 are wrong; the Au/Pd/ZIF-8 does not exhibit high hydrogenation rate at low H₂ concentration. This is crucial since all of the following discussions hinge on the conclusions derived from the EXAFS analysis. On the other hand, if the Figure is wrong, then the $K_{r100\%}/K_{r4\%}$ ratios reported in Figure 2e and cited in page 9 lines 168-169 are also wrong.

We thank the referee for his/her insightful comments and suggestions. We have modified our typo in the manuscript. k_R represents the hydrogenation rate of PdH_x, but is affected by many factors, such as surface defects and charge transfer interactions (*J. Catal.* **43**, 376-379 (1976)).

For Au/Pd and Au/Pd/ZIF-8, it is complicated to interpret the difference in k_R of the two samples. However, in the same sample, k_R with different hydrogen concentrations can be directly compared. According to A. Borgschulte's research, the relationship between the palladium hydride hydrogenation rate (R) and the applied pressure (p) is established (*Phys. Rev. B* **78**, 094106 (2008)). It was acknowledged that the hydrogenation rate (R) is proportional to the surface hydrogen concentration according to the Fick's law of diffusion. The relationship between R and p can be simplified as follows:

$$\frac{R_1}{R_2} = \sqrt{\frac{p_1}{p_2}}$$

As mentioned above, $R \propto k_r, k_{CN}$ and $\frac{p_{100\%}}{p_{4\%}} = 5$. For AuPd, $\frac{k_{R100\%}}{k_{R4\%}} = 5.53$, which is in good agreement with the theory. However, $\frac{k_{R100\%}}{k_{R4\%}} = 1.91$ for AuPd@ZIF-8 is quite different from the theoretical results. The results mentioned above imply that AuPd@ZIF-8 has a higher apparent hydrogen pressure or surface hydrogen concentration under 4% H_2 conditions.

5. In page 10 line 185: “suggesting the reduced electron density of Pd in Au/Pd/ZIF-8. This indicates that the 4d-electrons of Pd slightly transfer to ZIF-8” This shall be easy and straightforward analysis by DFT. It would be great if the author can prove this charge transfer by DFT. Also, The ZIF is also complicated. Will this e of Pd go to which part of ZIF?

We thank the referee for his/her insightful comments and suggestions. Upon conducting Bader charge analysis, it was revealed that ZIF-8 received $0.3 e^-$, indicative of a slight transfer of 4d-electrons from Pd to ZIF-8. To further elucidate the direction and distribution of electron transfer, atomic color map was constructed based on the findings from the Bader charge analysis. In this color representation, blue color corresponds to an electron gain, while red color indicates an electron loss. Insights drawn from the atomic color map suggest that the transfer of electrons occurs predominantly from Pd to N within ZIF-8.

Fig. S11 | (a) Bader analysis of Pd-ZIF-8. (b) Atomic coloring diagram of Pd-ZIF-8. Blue color corresponds to an electron gain, while red color indicates an electron loss.

6. In page 10 lines 189-195, “the dynamic process of H_{ad} atoms” is confusing. H_{ad} atoms itself is not a process. Can the specify here what type of dynamic process occurring and why is it “dynamic”?

We thank the referee for his/her insightful comments and suggestions. The dynamic process of H_{ad} atoms includes the H_{ad} atom spillover to the the ZIF-8 coating that reduces the H_{ad} concentration, as well as the H_{ad} atom migration from the ZIF-8 coating to the Pd surface that increases the H_{ad} concentration. In this way, the dynamic processes of H_{ad} atoms in the interface ZIF-8 regime would alter the H_{ad} concentration at the Pd–ZIF-8 interface.

7. In Figure 4, the resolution of the images is too low. In connection to this, in Figure 4b, where the H supposed goes from ZIF down to Pd, it is confusing why the x-axis of the plot indicates that the hydrogen atom is moving up (z coordinate is increasing). In connection to this, the following discussions in page 14 are confusing:

a. Line 260: “In comparison, the potential barrier of hydrogen spillover from Pd (100) surface to ZIF-8 is 1.0 eV (Fig. 4a)”. However, looking at Figure 4a, the barrier of migration from Pd to ZIF seems to be only 0.2 eV (calculating the difference between the peak and the first local minima: $1.0 - 0.8 = 0.2$ eV)

b. Line 264: “In the meantime, we notice that the H_{ad} in ZIF-8 has a very low barrier to migrate back to the Pd surface (Fig. 4b).” Please cite the value of the barrier for the reverse reaction. Also can the authors comment on why the forward and reverse migration of H spillover between Pd and ZIF have different barriers? Do they proceed via different mechanisms?

We thank the referee for his/her insightful comments and suggestions. We have adjusted the resolution of the image in Figure 4 in the main text. The X-axis in Figure 4(a) primarily stems from the constrained molecular dynamics in which we set the distance of C–H as the collective variable (CV) from Pd to ZIF-8 (Supplementary Fig. 22). This decrease in C–H distance has been implemented to simulate the process of hydrogen spillover.

In Figure 4(b), we have established the Z-coordinate of the H atom as the CV value (Supplementary Fig. 22), using its diminishing process to simulate the energy barrier for the hydrogen atom moving from ZIF-8 to Pd.

(a) Consequently, the potential barrier of hydrogen spillover from the Pd (100) surface to ZIF-8 is 1.0 eV, not 0.2 eV (Fig. 4a).

(b) Similarly, the energy barrier in Figure 4b should be viewed from right to left. Hence, the hydrogen atom within ZIF-8 has a very low barrier to migrate back to the Pd surface (Fig. 4b).

When the concentration of hydrogen atoms is high, spillover from the Pd surface to the MOF occurs. However, due to the adsorption of hydrogen atoms by Pd, this spillover has to overcome a chemisorption energy barrier of approximately 1.0 eV. In contrast, in the process of spillover from the MOF to Pd, ZIF-8 exhibits physisorption so that its desorption requires almost no energy barrier. Moreover, due to the chemisorption between Pd and hydrogen, the reflux process may be exothermic (Fig. 4b). Therefore, the mechanisms of the two processes are slightly different: the former needs to overcome a chemisorption energy barrier, while the latter is merely a process of physisorption desorption.

Fig. 4 | Theoretical calculations for reversible hydrogen spillover in Pd/ZIF-8 structures. **a–b**, Free energy profiles of hydrogen atom spillover **(a)** from Pd to ZIF-8 (the X-axis represents the distance of C–H) and **(b)** from ZIF-8 back to Pd (the X-axis represents the Z-coordinate of the hydrogen atom). The position of hydrogenation is illustrated by the schematics. **c**, Schematic demonstrating the methods used to examine hydrogen spillover.

8. With regards to the hydrogenation of organic compounds discussed in pages 15-16, can the authors elaborate on the envisioned mechanism of such reactions? The HSPE is provided as the reason for the improved activity of Pd/ZIF-8 compared to Pd. However, the interface between Pd and ZIF-8 seems to be inaccessible to reactants or adsorbates with significant size such as the organic compounds used. Thus, it is not clear how organic compounds would bind to the Pd/ZIF-8 catalyst and how hydrogen spillover participates in the reaction. In addition, if a mechanism similar to that depicted in Fig. 3d is assumed, should not the requirement of H_{ad} diffusion across the ZIF-8 layer, at least in principle, slow down the hydrogenation of organic compounds compared to the bare Pd?

We thank the referee for his/her insightful comment and suggestion. The hydrogenation of alkynes on Pd surface has been well studied by many researchers. Alkyne molecules adsorbed on the Pd surface are hydrogenated by H atoms. The rate-limiting step of alkyne hydrogenation is usually assumed to be the first H-addition step (*Science* **320**, 86-89 (2008)). As such, the H_{ad}

concentration is a key factor for alkyne hydrogenation. In addition, as a series of commonly used catalysis, plenty of literature show that reactants can migrate through the pores of MOF and reach the metal surface (*Angew. Chem.* **55**, 3685-3689 (2016)). The large pore size for ZIF-8 is 11.6 Å in diameter, and the small aperture is 3.4 Å in diameter. Meanwhile, the molecular sizes of DAC, DMAD and MBY are ca. 3.7 Å, ca. 3.5 Å and ca. 4.0 Å. In this way, the reactants could migrate through the ZIF-8 coating. According to the mechanism mentioned before, a high H_{ad} concentration would accelerate the hydrogenation reaction and lead to high hydrogenation conversion. On the basis of the results of quick-EXAFS, the ZIF-8 coating could alter the H_{ad} concentration at the Pd-ZIF-8 interface. In particular, ZIF-8 coating increases the H_{ad} concentration at 4% H_2 , which would accelerate the hydrogenation of organic compounds. The anomalously high efficiency/selectivity performance at a 4% hydrogen concentration prompts the discovery of the reversible hydrogen spillover phenomenon.

REVIEWER COMMENTS

Reviewer #1 (Remarks to the Author):

In this revised manuscript, the authors have answered all the comments of this reviewer, and the quality of the manuscript has been improved. The reviewer is ready to go with other two reviewers if they are also satisfied with the revisions.

Reviewer #2 (Remarks to the Author):

The revised version of the manuscript does not properly address some of the important comments of this Reviewer, and it is not suitable for publication in Nature Communications. Below are the detailed comments:

(1) The authors indicate that “for Au/Pd and Au/Pd/ZIF-8, it is complicated to interpret the difference in kR of the two samples. However, in the same sample, kR with different hydrogen concentrations can be directly compared”. They calculate the ratio of $kR_{100\%}/kR_{4\%}$ and obtain different ratio over AuPd@ZIF-8 and Au/Pd, and thus they conclude that AuPd@ZIF-8 has a higher apparent hydrogen pressure or surface hydrogen concentration under 4% H₂ conditions. However, the above inference is unconvincing and lack of logic. Why the lower ratio of $kR_{100\%}/kR_{4\%}$ implies an extraordinarily high rate at low H₂ concentration? The ratio of $kR_{100\%}/kR_{4\%}$ only shows the relative relation of hydrogenation rate under different pressure. We can conclude that the hydrogenation rate of AuPd@ZIF-8 is insensitive to H₂ pressure, but we cannot conclude the higher apparent hydrogen pressure or surface hydrogen concentration over AuPd@ZIF-8. It should be noted that absolute value of kR over AuPd@ZIF-8 is lower than that over AuPd.

The authors indicate that “The relationship between R and P can be simplified as follows:

$R_1/R_2=(P_1/P_2)^{1/2}$. As mentioned above, $R \propto kr$, kCN and $P_{100\%}/P_{4\%} = 5$. For AuPd, $kR_{100\%}/kR_{4\%} = 5.53$, which is in good agreement with the theory. However, $kR_{100\%}/kR_{4\%} = 1.91$ for AuPd@ZIF-8 is quite different from the theoretical results”. This statement seems incorrect. Please note that $R_1/R_2 \propto (P_1/P_2)^{1/2}$. According the authors statement, $kR_{100\%}/kR_{4\%} \propto (P_{100\%}/P_{4\%})^{1/2} = (5)^{1/2} = 2.23$, which indicated that AuPd@ZIF-8 ($kR_{100\%}/kR_{4\%} = 1.91$) is more agree with the theory?

(2) The authors collect the SEM and AFM images to confirm the continuous and compact coatings. However, the pores (i.e. mesoporous) cannot be excluded by the images of SEM and AFM. Physical absorption or other characterizations should be applied to evaluate the pores size distribution of ZIF-8 layer. Another comment that “How to confirm each silicon wafer contains the same active site of Pd and Au?” has not been addressed. pNTP hydrogenation rate should be normalization by the exposed active sites (TOF could be calculated).

Reviewer #3 (Remarks to the Author):

The innovative experiments and catalyst design reported in this study are interesting. However, the logical inconsistencies, incomplete description of some results, and incomplete evidence for major conclusions make it unworthy to be published in the esteemed journal Nature Communications. The following concerns, which were either already raised in the first review, or arising from a new understanding of the revised manuscript, need to be addressed before considering publication.

- 1) In Figure 1b, the use of the term “hydrogen spillover” was correctly changed to “hydrogen migration” within the figure but the caption was not changed.
- 2) On page 8 line 156, based on the intext descriptions of PdH_α and PdH_β , it seems that the only difference is PdH_β corresponds to H-saturated Pd. Can the author explain the purpose of distinguishing two H by alpha and beta?
- 3) On page 9 line 171, the typo error regarding the k_r values has been corrected. Moreover, the clarity of the section was improved in the revision. It is now clear that the authors compare $k_{R100\%}/k_{R4\%}$ ratios for Au/Pd and Au/Pd/ZIF-8, not the individual k_R values.

However, based on the $k_{R100\%}/k_{R4\%}$ ratios of 5.5 and 1.9 for Au/Pd and Au/Pd/ZIF-8, it is not apparent that “under 4% H_2 condition, Au/Pd/ZIF-8 has a higher hydrogen concentration on Pd surface” (as the authors wrote in page 9, line 185-186). This conclusion presumes that the hydrogenation rate at 100% H_2 is equal between Au/Pd and Au/Pd/ZIF-8. However, the $k_{R100\%}$ value of Au/Pd/ZIF-8 ($1.7 \times 10^{-4} \text{ \AA s}^{-1}$) is less than 25% that of compared to Au/Pd ($4.2 \times 10^{-4} \text{ \AA s}^{-1}$). I understand that the different chemical environment between Au/Pd/ZIF-8 and Au/Pd makes it difficult to compare individual $k_{R100\%}$ values. However, given such a glaring difference in the individual $k_{R100\%}$ values, it is easier to imagine that the presence of ZIF-8 inhibits the hydrogenation of the Pd surface in a way that scales with % H_2 , lowering the $k_{R100\%}/k_{R4\%}$ ratio for Au/Pd/ZIF-8, versus Au/Pd. Without ruling out this possibility, the Quick-EXAFS data alone does not decisively prove that hydrogen concentration is higher on Au/Pd/ZIF-8 versus Au/Pd.

I would suggest performing calculations to supplement the results of the experiment. For instance, the H_{ad} adsorption energy can be compared on Au/Pd/ZIF-8 versus Au/Pd, for high and low H_{ad} coverage. If the authors would like to further understand the kinetic properties, activation energies for dissociation and diffusion might also be calculated to understand the kinetic properties, such as reaction rates.

- 4) In Figure 4, the x-axes used for Figures 4a and 4b are now coherent and described appropriately. However, looking at the details of the Figures 4a and 4b leads to some questions in the cAIMD calculations described in page 4-5 of the SI and the consistency of the results.
 - a. The total AIMD timescale used to resolve the reaction pathways are not disclosed.
 - b. It is not explained why different constrained variables (CV) had to be used to simulate forward (Figure 4a using C-H distance) and reverse spillover (Figure 4b using z-axis of spillover H).
 - c. The curve in Figure 4b is not smooth, which might indicate that the AIMD timestep is not small enough to resolve the reaction pathway. Systems with highly mobile hydrogen atoms commonly require timestep of 0.5 fs or less.
 - d. I think that reporting the two pathways (in Figures 4a and 4b) using the same variable in the x-axis, regardless of the CV used, is better for easier comparison and consistency. I would also

suggest marking in the figure how the barriers are calculated. For example, a vertical double-ended arrow with the value of the activation energy could be drawn.

- e. Figures 4a and 4b depict the same elementary step in opposite directions. The authors need to explain why the forward and reverse reactions show distinct energy profiles. In Figure 4a, the spillover of hydrogen from Pd to ZIF8 is endothermic by about 0.8 eV. In Figure 4b, the reverse spillover from ZIF8 to Pd is exothermic by approximately 0.4 eV and lacks a noticeable barrier. These differences violate the principle of microscopic reversibility in chemical reactions, suggesting either distinct mechanistic pathways or intermediates were resolved for the two figures. Consequently, comparing energy barriers between Figures 4a and 4b, which hydrogen migration to Pd was concluded, requires careful reconsideration.
- 5) It would help the manuscript if part or the entirety of the mechanisms for the catalytic hydrogenation of unsaturated hydrocarbons, which the authors described in their response letter, be included in the manuscript.
 - 6) I might suggest the authors include the parts of "original manuscript" and "revised manuscript", corresponding to that question and response in the response letter in the future. So it would be much faster to understand if the authors modify the corresponding paragraphs in revised manuscript or SI.

Reviewer #1 (Remarks to the Author):

In this revised manuscript, the authors have answered all the comments of this reviewer, and the quality of the manuscript has been improved. The reviewer is ready to go with other two reviewers if they are also satisfied with the revisions.

We really appreciate the referee's highly positive evaluation of our work.

Reviewer #2 (Remarks to the Author):

The revised version of the manuscript does not properly address some of the important comments of this Reviewer, and it is not suitable for publication in Nature Communications. Below are the detailed comments:

We would like to express our gratitude to the referee for his/her insightful comments and suggestions, which have greatly contributed to enhancing the quality of our manuscript. We have diligently incorporated these suggestions and sincerely believe that our revisions have effectively addressed the concerns raised by the referee.

(1) The authors indicate that “for Au/Pd and Au/Pd/ZIF-8, it is complicated to interpret the difference in k_R of the two samples. However, in the same sample, k_R with different hydrogen concentrations can be directly compared”. They calculate the ratio of $k_{R100\%}/k_{R4\%}$ and obtain different ratio over AuPd@ZIF-8 and Au/Pd, and thus they conclude that AuPd@ZIF-8 has a higher apparent hydrogen pressure or surface hydrogen concentration under 4% H_2 conditions. However, the above inference is unconvincing and lack of logic. Why the lower ratio of $k_{R100\%}/k_{R4\%}$ implies an extraordinarily high rate at low H_2 concentration? The ratio of $k_{R100\%}/k_{R4\%}$ only shows the relative relation of hydrogenation rate under different pressure. We can conclude that the hydrogenation rate of AuPd@ZIF-8 is insensitive to H_2 pressure, but we cannot conclude the higher apparent hydrogen pressure or surface hydrogen concentration over AuPd@ZIF-8. It should be noted that absolute value of k_R over AuPd@ZIF-8 is lower than that over AuPd.

The authors indicate that “The relationship between R and P can be simplified as follows: $R_1/R_2=(P_1/P_2)^{1/2}$. As mentioned above, $R \propto k_r$, k_{CN} and $P_{100\%}/P_{4\%} = 5$. For AuPd, $k_{R100\%}/k_{R4\%} = 5.53$, which is in good agreement with the theory. However, $k_{R100\%}/k_{R4\%} = 1.91$ for AuPd@ZIF-8 is quite different from the theoretical results”. This statement seems incorrect. Please note that $R_1/R_2 \propto (P_1/P_2)^{1/2}$. According the authors statement, $k_{R100\%}/k_{R4\%} \propto (P_{100\%}/P_{4\%})^{1/2} = (5)^{1/2} = 2.23$, which indicated that AuPd@ZIF-8 ($k_{R100\%}/k_{R4\%} = 1.91$) is more agree with the theory?

We thank the referee for providing such insightful comments and suggestions, which have been invaluable in refining our manuscript and enhancing its logical coherence. To further illustrate the surface hydrogen concentration of Au/Pd/ZIF-8 and Au/Pd, we have conducted calculations of the adsorption energy of hydrogen atoms (H_{ad}) for both high ($\theta=1.00$) and low H_{ad} coverage ($\theta=0.03$). For Au/Pd/ZIF-8, the H_{ad} adsorption energy is -0.524 eV for $\theta=1.00$ and -0.819 eV for $\theta=0.03$. In comparison, for Au/Pd, the H_{ad} adsorption energy is -0.387 eV for $\theta=1.00$ and -0.517 eV for $\theta=0.03$ (Supplementary Fig. 10). The higher H_{ad} adsorption energy indicates that hydrogen atoms are adsorbed more easily on the Pd surface, resulting in a higher concentration of hydrogen atoms on the Pd surface in Au/Pd/ZIF-8 compared to Au/Pd under varying H_2 conditions (*Surf Sci.* **99**, 320-340 (1980); *Surf Sci.* **401**, 344-354 (1998); *Angew. Chem. Int. Ed.* **58**, 14534-14538 (2019); *Surf Sci.* **307-309**, 76-81 (1994); *Surf Sci.* **41**, 435-446 (1974)).

Furthermore, the potential barrier of hydrogen atoms passing through Pd (100) surface to octahedral sites is 0.64 eV for Au/Pd/ZIF-8 and 0.37 eV for Au/Pd, which illustrates that hydrogen atoms are more difficult to penetrate into Pd lattice in Au/Pd/ZIF-8 (Supplementary Fig.11) (*Int. J. Mater. Res.* **108**, 785-790 (2017)). Despite the higher hydrogen concentration on the Pd surface in Au/Pd/ZIF-8, the higher potential barrier may explain why the absolute value of k_R for Au/Pd/ZIF-8 is lower than that for Au/Pd. Therefore, the lower k_R value and the higher hydrogen concentration on the Pd surface of Au/Pd/ZIF-8 are not contradictory. According to the suggestion, we have addressed the state that ZIF-8 layers enable an insensitive property for H_2 concentration.

Regarding the second question, we thank the referee for pointing out this typo-error in the formula used in the manuscript. The accurate expression is indeed $\sqrt{\frac{p_{100\%}}{p_{4\%}}} = 5$. In this way, for AuPd, $\frac{k_{R100\%}}{k_{R4\%}} = 5.53$, which aligns well with the theory. However, $\frac{k_{R100\%}}{k_{R4\%}} = 1.91$ for Au/Pd/ZIF-8 indicates a significant deviation from the theoretical results. The results mentioned above imply that the hydrogenation rate of Au/Pd is proportional to the $p_{H_2}^{\frac{1}{2}}$ and the hydrogenation rate of Au/Pd/ZIF-8 deviates from the Fick's law which is not proportional to the $p_{H_2}^{\frac{1}{2}}$.

Supplementary Fig. 10 | Structures of PdH and PdH-ZIF-8 from top view (left) and side view (right) with H_{ad} coverage (θ) of 0.03 and 1.00. (a) Structure of PdH $_{\theta=0.03}$. (b) Structure of PdH $_{\theta=1.00}$. (c) Structure of PdH $_{\theta=0.03}$ -ZIF-8. (d) Structure of PdH $_{\theta=1.00}$ -ZIF-8. Pd: golden; H: grey; Zn: green; C: light blue; N: navy blue.

(2) The authors collect the SEM and AFM images to confirm the continuous and compact

coatings. However, the pores (i.e., mesoporous) cannot be excluded by the images of SEM and AFM. Physical absorption or other characterizations should be applied to evaluate the pores size distribution of ZIF-8 layer. Another comment that “How to confirm each silicon wafer contains the same active site of Pd and Au?” has not been addressed. pNTP hydrogenation rate should be normalization by the exposed active sites (TOF could be calculated).

We deeply appreciate the referee’s insightful comments and suggestions. In order to investigate the potential presence of mesopores, we have conducted N₂ adsorption isotherm measurements on the ZIF-8 layer at 77 K using a volumetric adsorption apparatus. The sample with ZIF-8 layer can be obtained by scraping the Pd/ZIF-8 substrates. As depicted in Supplementary Fig. 16, the pore size is 30 Å, falling within the range characteristic of mesopores. Thus, the pores in the ZIF-8 layer are mesopores. However, the pore size of ZIF-8 is much smaller than Au particle size (ca. 45 nm), and the adsorption isotherm exhibits similarities with prior studies (*J. Phys. Chem. Lett.* **3**, 1159–1164 (2012); *J. Am. Chem. Soc.* **133**, 8900-8902 (2011)). To confirm that each silicon wafer contains the same active sites of Pd and Au, a combination of SEM, AFM, Raman imaging and *in situ* SERS characterization techniques have been employed. Based on SEM images, AFM analysis and particle size distribution data of Au particles (Supplementary Fig. 21), we can find that the Au particles are well-dispersed, with an average size of approximately 45 nm. Raman spectra intensities are consistent across different samples, indicating uniform active sites of Au (Supplementary Fig. 25).

Given that the Pd layer is covered by varying thicknesses of ZnO, it is imperative to remove the ZnO layer before further measurements. To this end, we have etched the ZnO layer with a 0.1 M HCl aqueous solution overnight. Subsequently, *in situ* SERS characterization has been employed to confirm the comparability of the Pd layer across different samples. As demonstrated in Fig. R1, the results of *in situ* SERS characterization provide evidence that each silicon wafer contains similar active sites of Pd and Au, as evidenced by their parallel pNTP hydrogenation rates under identical condition. The computation procedure of the TOF of pNTP hydrogenation is summarized as follows:

According to Van Hardeveld’s study, cuboctahedron is approximately equivalent to the sphere as for nanoparticles (*Surf Sci.* **15**, 189-230 (1969)). Thus, the total number of atoms (N_T) can be defined as follows:

$$N_T = 16m^3 - 33m^2 + 24m - 6$$

The number of surface atoms (N_S):

$$N_S = 30m^2 - 60m + 32$$

where m represents the number of atoms lying on an equivalent edge (corner atoms included). As such, the surface ratio of Au atoms:

$$\text{ratio}_S = \frac{N_S}{N_T}$$

On the basis of geometric principle,

$$m = \frac{d_{Au \text{ particle}}}{\sqrt{10} \times 2r_{Au \text{ atom}}}$$

where $d_{Au\ particle}$ represents the diameter of Au particle and $r_{Au\ atom}$ represents the section radius of Au atom. The $d_{Au\ particle}$ is *ca.* 45.0 nm and $r_{Au\ atom}$ is *ca.* 0.144 nm. In this case, the ratios can be calculated to be about 0.0380. The TOF can be calculated by the following equation:

$$TOF = \frac{mol_{product}}{ratio_S \times mol_{total\ Au} \times time}$$

where $mol_{product}$ is the number of pATP. TOF values can be calculated for cases with less than 20% conversion rate, where the substrate is in excess of the balanced endpoint. The results of TOF for different samples can be found in Supplementary Table 2. TOF of pNTP hydrogenation of different samples is $2.62 \times 10^4\ h^{-1}$ for Pd/Au, $1.42 \times 10^4\ h^{-1}$ for Pd/ZIF-8-20 nm/Au, $6.66 \times 10^3\ h^{-1}$ for Pd/ZIF-8-40 nm/Au, $3.35 \times 10^3\ h^{-1}$ for Pd/ZIF-8-60 nm/Au and $0\ h^{-1}$ for Pd/ZIF-8-120 nm/Au. The decrease in TOF with an increase in the thickness of the ZIF-8 layer implies that the hydrogenation of pNTP relies on the hydrogen spillover which supplies the hydrogen atoms from Pd surface.

Supplementary Fig. 16 | (a) Pore size distribution and (b) N₂ adsorption-desorption isotherms of ZIF-8.

Supplementary Fig. 21 | SEM images, AFM analysis ($2 \times 2 \mu\text{m}$) and particle size distribution of Au particles of (a) Pd/Au, (b) Pd/ZIF-8-20 nm/Au, (c) Pd/ZIF-8-40 nm/Au, (d) Pd/ZIF-8-60 nm/Au, and (e) Pd/ZIF-8-120 nm/Au.

Supplementary Fig. 25 | Raman imaging of Au particles of (a) Pd/Au, (b) Pd/ZIF-8-20 nm/Au, (c) Pd/ZIF-8-40 nm/Au, (d) Pd/ZIF-8-60 nm/Au, and (e) Pd/ZIF-8-120 nm/Au. The Raman map uses the peak area from 1300 to 1370 cm^{-1} for image formation.

Fig. R1 | *In situ* SERS characterization. (a) *In situ* SERS spectra of pNTP hydrogenation on Pd/Au. (b) Time-dependent Raman peak intensity for the nitro group of pNTP on Pd/Au-ZnO- x nm. The ZnO layer with different thicknesses were etched by 0.1 M HCl aqueous solution.

Supplementary Table 2 | TOF of pNTP hydrogenation of different samples.

Samples	Time (20% conversion)/h	TOF /h ⁻¹
Pd/Au	9.12×10^{-3}	2.62×10^4
Pd/ZIF-8-20 nm/Au	1.68×10^{-2}	1.42×10^4
Pd/ZIF-8-40 nm/Au	3.59×10^{-2}	6.66×10^3
Pd/ZIF-8-60 nm/Au	7.14×10^{-2}	3.35×10^3
Pd/ZIF-8-120 nm/Au	∞	0

Reviewer #3 (Remarks to the Author):

The innovative experiments and catalyst design reported in this study are interesting. However, the logical inconsistencies, incomplete description of some results, and incomplete evidence for major conclusions make it unworthy to be published in the esteemed journal Nature Communications. The following concerns, which were either already raised in the first review, or arising from a new understanding of the revised manuscript, need to be addressed before considering publication.

We really appreciate the referee's insightful comments and suggestions to help us significantly improve the quality of our manuscript. We have carefully revised the manuscript and sincerely hope that our revisions have satisfactorily addressed the referee's concerns.

1) *In Figure 1b, the use of the term "hydrogen spillover" was correctly changed to "hydrogen migration" within the figure but the caption was not changed.*

We thank the referee for bringing it to our attention. We have duly amended the caption of Figure 1b in the manuscript.

2) *On page 8 line 156, based on the intext descriptions of PdH_α and PdH_β, it seems that the only difference is PdH_β corresponds to H-saturated Pd. Can the author explain the purpose of distinguishing two H by alpha and beta?*

We thank the referee for his/her thoughtful comments. As demonstrated in Fig. 2d, the relationship between the bonding distance of Pd–Pd and time can be divided into two parts: linear part and nonlinear part. The linear expansion of the Pd lattice is directly proportional to the number of hydrogen atoms (representing the occupation of octahedral sites within the Pd lattice) within the range of PdH_α and PdH_β. It is only within the range of PdH_{α+β} that the Pd lattice expansion becomes nonlinear (*Hydrogen in Metals I: Basic Properties* (eds Georg Alefeld & Johann Völkl) 53-74 (Springer Berlin Heidelberg, 1978); *Solid State Phenom.* **73-75**, 366-422 (2000)). To elucidate the relationship between the bonding distance of Pd–Pd and time, we differentiate the PdH_x states into PdH_α, PdH_β and PdH_{α+β} regions.

3) *On page 9 line 171, the typo error regarding the K_r values has been corrected. Moreover, the clarity of the section was improved in the revision. It is now clear that the authors compare k_{R100%}/k_{R4%} ratios for Au/Pd and Au/Pd/ZIF-8, not the individual k_R values. However, based on the k_{R100%}/k_{R4%} ratios of 5.5 and 1.9 for Au/Pd and Au/Pd/ZIF-8, it is not apparent that "under 4% H₂ condition, Au/Pd/ZIF-8 has a higher hydrogen concentration on Pd surface" (as the authors wrote in page 9, line 185-186). This conclusion presumes that the hydrogenation rate at 100% H₂ is equal between Au/Pd and Au/Pd/ZIF-8. However, the k_{R100%} value of Au/Pd/ZIF-8 (1.7 × 10⁻⁴ Å s⁻¹) is less than 25% that of compared to Au/Pd (4.2 × 10⁻⁴*

\AA s^{-1}). I understand that the different chemical environment between Au/Pd/ZIF-8 and Au/Pd makes it difficult to compare individual $k_{R100\%}$ values. However, given such a glaring difference in the individual $k_{R100\%}$ values, it is easier to imagine that the presence of ZIF-8 inhibits the hydrogenation of the Pd surface in a way that scales with %H₂, lowering the $k_{R100\%}/k_{R4\%}$ ratio for Au/Pd/ZIF-8, versus Au/Pd. Without ruling out this possibility, the Quick-EXAFS data alone does not decisively prove that hydrogen concentration is higher on Au/Pd/ZIF-8 versus Au/Pd. I would suggest performing calculations to supplement the results of the experiment. For instance, the H_{ad} adsorption energy can be compared on Au/Pd/ZIF-8 versus Au/Pd, for high and low H_{ad} coverage. If the authors would like to further understand the kinetic properties, activation energies for dissociation and diffusion might also be calculated to understand the kinetic properties, such as reaction rates.

We thank the referee for his/her insightful comment and suggestion. As the referee pointed out, despite hydrogen concentration on Pd surface, many other factors, such as ZIF-8 inhibition of palladium hydrogenation, would lower the $k_{R100\%}/k_{R4\%}$ ratio. According to the suggestion, we have addressed this issue in the revised manuscript. To provide further insight into the surface hydrogen concentration, we have conducted calculations of the H_{ad} adsorption energy for both high ($\theta=1.00$) and low H_{ad} coverage ($\theta=0.03$). The computational formula is $E_{ad} = (E_{tot} - E_{slab} - \frac{1}{2}nE_{H_2})/n$, where E_{tot} and E_{slab} represent the energy of structures with or without hydrogen atoms, respectively, and n represents the number of hydrogen atoms. For Au/Pd/ZIF-8, the H_{ad} adsorption energy is -0.524 eV for $\theta=1.00$ and -0.819 eV for $\theta=0.03$. In comparison, for Au/Pd, the H_{ad} adsorption energy is -0.387 eV for $\theta=1.00$ and -0.517 eV for $\theta=0.03$ (Supplementary Fig. 10). The higher H_{ad} adsorption energy indicates that hydrogen atoms are adsorbed more easily on the Pd surface, resulting in a higher concentration of hydrogen atoms on the Pd surface in Au/Pd/ZIF-8 compared to Au/Pd under varying H₂ conditions (*Surf Sci.* **99**, 320-340 (1980); *Surf Sci.* **401**, 344-354 (1998); *Angew. Chem. Int. Ed.* **58**, 14534-14538 (2019); *Surf Sci.* **307-309**, 76-81 (1994); *Surf Sci.* **41**, 435-446 (1974)). Furthermore, the potential barrier of hydrogen atoms passing through Pd (100) surface to octahedral sites is 0.64 eV for Au/Pd/ZIF-8 and 0.37 eV for Au/Pd, which illustrates that hydrogen atoms are more difficult to penetrate into Pd lattice in Au/Pd/ZIF-8 (Supplementary Fig.11) (*Int. J. Mater. Res.* **108**, 785-790 (2017)). Combining higher H_{ad} adsorption energy and higher potential barrier of Au/Pd/ZIF-8, it is reasonable to conclude that Au/Pd/ZIF-8 shows higher hydrogen concentration and lower hydrogenation rate.

Due to the complexity of the Pd/ZIF-8 system (more than 200 atoms in system), it is understandable that calculating kinetic properties, activation energies for dissociation and diffusion in such system would be time-consuming, potentially taking months or even years. Nevertheless, this research direction warrants further investigation in our future studies.

Supplementary Fig. 10 | Structures of PdH and PdH-ZIF-8 from top view (left) and side view (right) with H_{ad} coverage (θ) of 0.03 and 1.00. (a) Structure of PdH $_{\theta=0.03}$. (b) Structure of PdH $_{\theta=1.00}$. (c) Structure of PdH $_{\theta=0.03}$ -ZIF-8. (d) Structure of PdH $_{\theta=1.00}$ -ZIF-8. Pd: golden; H: grey; Zn: green; C: light blue; N: navy blue.

4) In Figure 4, the x-axes used for Figures 4a and 4b are now coherent and described appropriately. However, looking at the details of the Figures 4a and 4b leads to some questions in the cAIMD calculations described in page 4-5 of the SI and the consistency of the results.

a. The total AIMD timescale used to resolve the reaction pathways are not disclosed.

We thank the referee for his/her insightful comments and suggestions. For the free energy profile of hydrogen spillover from Pd to ZIF-8, the collective variable (CV) (C-H distance) increment was set to 0.0004 Å and the simulation time was set to 6.5 ps. For the free energy profile of hydrogen spillover from ZIF-8 back to Pd, CV was set to 0.00003 and simulation time was set to 5 ps. The detailed CV settings are shown in Supplementary Fig. 27.

b. It is not explained why different constrained variables (CV) had to be used to simulate forward (Figure 4a using C-H distance) and reverse spillover (Figure 4b using z-axis of spillover H).

We appreciate the insightful comments provided by the referee. Figures 4a and 4b depict two distinct processes in a straightforward way: hydrogen spillover from the Pd surface to ZIF-8 under high surface hydrogen concentration, and hydrogen spillover from ZIF-8 to the Pd surface under low surface hydrogen concentration. It is worth pointing out that reversible migration of hydrogen does not occur at the same time and the same conditions. The prevalence

of either spillover process is dictated by the surface hydrogen concentration in the Pd/ZIF-8 system. It has been proven ineffective when using the same variable to represent both processes. Specifically, employing the same variable, C–H distance, in AIMD calculations for both processes raises questions about the practicality of spillover from the Pd surface to ZIF-8. In the case of spillover from ZIF-8 to the Pd surface, hydrogen atoms would migrate to ZIF-8 rather than the Pd layer, resulting in an increase in the C–H distance variable. To address this issue, we have chosen two appropriate CV values to accurately portray the hydrogen spillover in this system.

c. The curve in Figure 4b is not smooth, which might indicate that the AIMD timestep is not small enough to resolve the reaction pathway. Systems with highly mobile hydrogen atoms commonly require timestep of 0.5 fs or less.

We thank the referee for his/her valuable comments and suggestions. Typically, a timestep of 0.5 fs or less was used for systems with highly mobile hydrogen atoms, assuming an atomic mass of 1 for hydrogen atoms. However, in our system, the atomic mass of hydrogen atoms is 2 to reduce the calculation time which is common in AIMD calculation of complex system which contains numerous hydrogen atoms (*Proc. Natl. Acad. Sci.* **114**, 1795-1800 (2017); *J. Mater. Chem. A*, **9**, 23515-23521 (2021)).

Considering the results and insights from the aforementioned studies, it appears that the timestep of hydrogen atoms is not the primary factor contributing to the irregularity of the curve in Figure 4b. Instead, the choice of the collective variable (CV) may have a more significant impact. To enhance the smoothness of the curve in Figure 4b, a more refined value of CV has been employed to simulate the free energy profile of hydrogen spillover from ZIF-8 back to Pd. The CV was set to 0.00003, and the simulation time was set to 5 ps. The result is presented in Fig. 4b, illustrating that hydrogen in ZIF-8 has a very low barrier (0.1 eV) to migrate back to the Pd surface, which is consistent with the findings of previous AIMD calculations.

Fig. 4 | Theoretical calculations for reversible hydrogen spillover in Pd/ZIF-8 structures. **a–b**, Free energy profiles of hydrogen atom spillover **(a)** from Pd to ZIF-8 (the X-axis represents the distance of C–H) and **(b)** from ZIF-8 back to Pd (the X-axis represents the Z-coordinate of the hydrogen atom). The position of hydrogenation is illustrated by the schematics. **c**, Schematic demonstrating the methods used to examine hydrogen spillover.

d. I think that reporting the two pathways (in Figures 4a and 4b) using the same variable in the x-axis, regardless of the CV used, is better for easier comparison and consistency. I would also suggest marking in the figure how the barriers are calculated. For example, a vertical double ended arrow with the value of the activation energy could be drawn.

We thank the referee for his/her valuable comments and suggestions. While maintaining the same variable on the x-axis can enhance comparison and consistency, it may not effectively capture both processes, as elucidated in our response to Comment 4b. Given the distinct nature of hydrogen spillover processes from the Pd surface to ZIF-8 and from ZIF-8 to the Pd surface, involving different surface hydrogen concentrations, utilizing the same variable would introduce confusion. To address this concern, we have meticulously selected two appropriate collective variable (CV) values to accurately represent hydrogen spillover in this system.

Furthermore, to enhance clarity, we have incorporated vertical double-ended arrows in our manuscript (Fig. 4a and 4b) to denote the barriers. Specifically, the potential barrier for hydrogen spillover from the Pd surface to ZIF-8 is 1.0 eV, while the barrier for hydrogen spillover from ZIF-8 to the Pd surface is 0.1 eV.

Fig. 4 | Theoretical calculations for reversible hydrogen spillover in Pd/ZIF-8 structures. **a–b**, Free energy profiles of hydrogen atom spillover **(a)** from Pd to ZIF-8 (the X-axis represents the distance of C–H) and **(b)** from ZIF-8 back to Pd (the X-axis represents the Z-coordinate of the hydrogen atom). The position of hydrogenation is illustrated by the

schematics. **c**, Schematic demonstrating the methods used to examine hydrogen spillover.

e. Figures 4a and 4b depict the same elementary step in opposite directions. The authors need to explain why the forward and reverse reactions show distinct energy profiles. In Figure 4a, the spillover of hydrogen from Pd to ZIF-8 is endothermic by about 0.8 eV. In Figure 4b, the reverse spillover from ZIF-8 to Pd is exothermic by approximately 0.4 eV and lacks a noticeable barrier. These differences violate the principle of microscopic reversibility in chemical reactions, suggesting either distinct mechanistic pathways or intermediates were resolved for the two Figures. Consequently, comparing energy barriers between Figures 4a and 4b, which hydrogen migration to Pd was concluded, requires careful reconsideration.

We thank the referee for his/her insightful comments and suggestion. Figures 4a and 4b illustrate the distinct processes of hydrogen spillover from the Pd surface to ZIF-8 under high surface hydrogen concentration and hydrogen spillover from ZIF-8 to the Pd surface under low surface hydrogen concentration. It is worth pointing out that reversible migration of hydrogen does not occur at the same time and the same conditions. The prevalence of either spillover process is dictated by the surface hydrogen concentration in the Pd/ZIF-8 system. Additionally, it is well-established in the literature that the adsorption of hydrogen atoms on the Pd surface is characterized by chemisorption, whereas the adsorption of hydrogen atoms in ZIF-8 is governed by physisorption (*Nat. Mater.* **13**, 802-806 (2014); *Phys. Rev. Lett.* **104**, 236101 (2010); *Phys. Rev. Lett.* **42**, 456-458 (1979)).

Consequently, the adsorption mechanism of hydrogen atoms spilling over from the Pd surface to ZIF-8 undergoes a transition from chemisorption to physisorption. This process is endothermic, requiring approximately 1.0 eV to break the Pd–H bond. On the other hand, the migration from ZIF-8 to the Pd surface involves a change in the adsorption mechanism of hydrogen atoms from physisorption to chemisorption. This process is exothermic, releasing approximately 0.4 eV and forming the Pd–H bond. According to the suggestion, we have modified the statement to “reversible movement process of H atoms”.

5) It would help the manuscript if part or the entirety of the mechanisms for the catalytic hydrogenation of unsaturated hydrocarbons, which the authors described in their response letter, be included in the manuscript.

We thank the referee for his/her insightful comment and suggestion. The addition of Fig. 5d to illustrate the hydrogenation mechanism is appreciated. This enhances the clarity and completeness of the manuscript regarding the involved catalytic processes.

Fig. 5 | Catalytic performance of Au/Pd and Au/Pd/ZIF-8. **a**, Structures of four acetylenes— diethyl acetylenedicarboxylate (A), dimethyl acetylenedicarboxylate (B), 2-methyl-3-butyn-2-ol (C) and acetylene (D). **b–c**, TOF of acetylenes hydrogenation of Au/Pd and Au/Pd/ZIF-8 under irradiation (**b**) and heating (**c**) conditions. **d**, Mechanism for catalytic hydrogenation of unsaturated hydrocarbons. Reaction conditions of alkyne hydrogenation: catalyst (containing 0.1 mg Pd), substrate (DAC, MBY, or DMAD, 10 μ L), solvent (CH₃CN, 1 mL), 4% H₂ (101 kPa). Reaction conditions of acetylene hydrogenation: catalyst (containing 0.1 mg Pd), 1% C₂H₂ and 4% H₂ (101 kPa) and 1 mL deionized water. All the used gas is balanced by Ar. Light intensity: 100 mW/cm². Heating temperature: 50 $^{\circ}$ C.

6) I might suggest the authors include the parts of “original manuscript” and “revised manuscript”, corresponding to that question and response in the response letter in the future. So, it would be much faster to understand if the authors modify the corresponding paragraphs in revised manuscript or SI.

We thank the referee for his/her insightful comment and suggestion. We will provide both the original manuscript and revised manuscript to the referees in the future.

REVIEWER COMMENTS

Reviewer #2 (Remarks to the Author):

The authors addressed most of the review's comments. However, the most important comment in Q2 has not been adequately addressed, which is critical for the proposed concept of "reversible hydrogen spillover". This problem should be well addressed before the manuscript is accepted in Nature Communications.

In the first round R1, the reviewer indicated that "during the formation of ZIF-8 layer, large pores could also generate which makes the Pd layer could directly interact with pNTP and thus lead to the higher activity. The increased thickness of ZIF-8 could decrease the defect pores which leads to the decrease of activity", the additional N₂ adsorption isotherm measurements (provided by the authors in the second round R2) indicated that the ZIF-8 layer contains mesopores. Although the mesopore size is much smaller than Au particle size, these mesopores cannot be block but the Au particle (as shown in Supplementary Fig. 21), which means that pNTP can diffuse through the mesopores and be directly hydrogenated on the Pd sites. With the increase of layer thickness, the increased diffusion resistance leads to the decrease of hydrogenation rate in Fig. 3. That is, the pNTP hydrogenation could mainly directly occurred on the Pd surface, but not on the Au with assistance of hydrogen spillover. The discussion and conclusion based on Fig. 3 might be incorrect! The author ascribed the pNTP hydrogenation to the hydrogen spillover and indicated that "the hydrogen spillover can indeed take place from interface into ZIF-8 but with a penetration depth limit". However, the reaction rate tendency in Fig. 3 could also be ascribed to the diffusion resistance from the ZIF-8 layer for pNTP reactant.

Reviewer #3 (Remarks to the Author):

In this revision, the authors have satisfactorily answered the comments and greatly improved the manuscript. There are no further logical inconsistencies in this version of the manuscript, which may be published as it is. Nevertheless, I strongly recommend that Figure 4 and its caption be improved by emphasizing that Figures 4a and 4b are simulated at different hydrogen coverage of the Pd surface. This will prevent readers from confusing them as exactly reversible processes, and makes it easier to understand the rest of the discussions.

Reviewer #2 (Remarks to the Author):

The authors addressed most of the review's comments. However, the most important comment in Q2 has not been adequately addressed, which is critical for the proposed concept of "reversible hydrogen spillover". This problem should be well addressed before the manuscript is accepted in Nature Communications.

We would like to express our gratitude for the reviewer's invaluable insights, which have greatly contributed to enhancing the quality of our manuscript. We have diligently incorporated these suggestions and sincerely believe that our revisions have effectively addressed the concerns raised by the reviewer.

In the first round R1, the reviewer indicated that "during the formation of ZIF-8 layer, large pores could also generate which makes the Pd layer could directly interact with pNTP and thus lead to the higher activity. The increased thickness of ZIF-8 could decrease the defect pores which leads to the decrease of activity", the additional N₂ adsorption isotherm measurements (provided by the authors in the second round R2) indicated that the ZIF-8 layer contains mesopores. Although the mesopore size is much smaller than Au particle size, these mesopores cannot be block but the Au particle (as shown in Supplementary Fig. 21), which means that pNTP can diffuse through the mesopores and be directly hydrogenated on the Pd sites. With the increase of layer thickness, the increased diffusion resistance leads to the decrease of hydrogenation rate in Fig. 3. That is, the pNTP hydrogenation could mainly directly occurred on the Pd surface, but not on the Au with assistance of hydrogen spillover. The discussion and conclusion based on Fig. 3 might be incorrect! The author ascribed the pNTP hydrogenation to the hydrogen spillover and indicated that "the hydrogen spillover can indeed take place from interface into ZIF-8 but with a penetration depth limit". However, the reaction rate tendency in Fig. 3 could also be ascribed to the diffusion resistance from the ZIF-8 layer for pNTP reactant.

We would like to express our sincere appreciation to the reviewer for providing such insightful comments and suggestions. Considering that only Au surface can be detected by Raman spectroscopy (*Nat. Nanotechnol.* **15**, 922–926 (2020)), it is well accepted that the pNTP molecules are mainly adsorbed on Au nanoparticles over Pd/ZIF-8/Au composite structure. Consequently, if pNTP molecules can diffuse to the Pd surface, they firstly need to desorb from the Au surface. Here, we therefore evaluate the pNTP desorption ability from Au nanoparticles surface via pNTP-temperature programmed desorption (TPD) and Raman spectroscopy.

First of all, we note that the pNTP (boiling point: 281.9 °C) is difficult to desorb from the Au surface only with a prominent desorption peak around 400–500 °C (Fig. R1) after saturation adsorption, suggesting a strong interaction between pNTP molecules and Au surface. In addition, the Raman spectra of Au nanoparticles film (Fig. R2) exhibit a negligible change of pNTP peaks before and after washing by alcohol solvent, which further indicates a strong bonding between pNTP molecules and Au nanoparticles. Lastly, we therefore synthesized Au@SiO₂ nanoparticles with porous shells using Liz-Marzán's method (*Langmuir* **12**, 4329–4335 (1996)). It is well known that the porous nature of SiO₂ shell can allow pNTP molecules to migrate, as evidenced by the strong Raman signals of pNTP. Supplementary Figure 25b demonstrates that pNTP molecules cannot be hydrogenated on the referenced Pd/Au@SiO₂ substrate, illustrating that active hydrogen atoms are unable to migrate through the silica shells to facilitate the hydrogenation over Au.

In summary, we conclude that the pNTP molecules are strongly adsorbed on the Au nanoparticles over Pd/ZIF-8/Au composite structure. More importantly, such Pd-ZIF-8-Au interface facilitates hydrogen spillover, thus providing a source of active hydrogen for pNTP hydrogenation on the Au surface.

Fig. R1 | TPD characterization of Au nanoparticles after pNTP saturation adsorption.

Fig. R2 | Raman spectra of Au nanoparticles film with pNTP (a) before and (b) after washing by alcohol solvent.

Supplementary Fig. 25 | *In situ* SERS spectra of pNTP hydrogenation on (a) Pd/Au and (b) Pd/Au@SiO₂.

Reviewer #3 (Remarks to the Author):

In this revision, the authors have satisfactorily answered the comments and greatly improved the manuscript. There are no further logical inconsistencies in this version of the manuscript, which may be published as it is. Nevertheless, I strongly recommend that Figure 4 and its caption be improved by emphasizing that Figures 4a and 4b are simulated at different hydrogen coverage of the Pd surface. This will prevent readers from confusing them as exactly reversible processes, and makes it easier to understand the rest of the discussions.

We really appreciate the reviewer's insightful suggestions to help us significantly improve the quality of our manuscript. We have carefully revised the Figure 4 and its captions in the revised manuscript, and sincerely hope that our revisions have satisfactorily addressed the reviewer's concerns.

Fig. 4 | Theoretical calculations for reversible hydrogen spillover in Pd/ZIF-8 structures. a–b, Free energy profiles of hydrogen atom spillover (a) from Pd to ZIF-8 (the X-axis represents the distance of C–H, high hydrogen coverage) and (b) from ZIF-8 back to Pd (the X-axis represents the Z-coordinate of the hydrogen atom, low hydrogen coverage). The position of hydrogenation is illustrated by the schematics. **c,** Schematic demonstrating the methods used to examine hydrogen spillover.

REVIEWERS' COMMENTS

Reviewer #2 (Remarks to the Author):

The authors have now adequately addressed the issue I raised. The manuscript is now acceptable for publication in Nature Communications.

Reviewer #2 (Remarks to the Author):

The authors have now adequately addressed the issue I raised. The manuscript is now acceptable for publication in Nature Communications.

We thank the referee for his/her positive evaluation.